# A Decision-Language Model (DLM) for Dynamic Restless Multi-Armed Bandit Tasks in Public Health

**Nikhil Behari** *
MIT, Harvard University

**Edwin Zhang***
Harvard University

**Yunfan Zhao**
GE Healthcare, Harvard University

**Aparna Taneja**
Google

**Dheeraj Nagaraj**
Google

**Milind Tambe**
Harvard University, Google

## Abstract

Restless multi-armed bandits (RMAB) have demonstrated success in optimizing resource allocation for large beneficiary populations in public health settings. Unfortunately, RMAB models lack flexibility to adapt to evolving public health policy priorities. Concurrently, Large Language Models (LLMs) have emerged as adept automated planners across domains of robotic control and navigation. In this paper, we propose a Decision Language Model (DLM) for RMABs, enabling dynamic fine-tuning of RMAB policies in public health settings using human-language commands. We propose using LLMs as automated planners to (1) interpret human policy preference prompts, (2) propose reward functions as code for a multi-agent RMAB environment, and (3) iterate on the generated reward functions using feedback from grounded RMAB simulations. We illustrate the application of DLM in collaboration with ARMMAN, an India-based non-profit promoting preventative care for pregnant mothers, that currently relies on RMAB policies to optimally allocate health worker calls to low-resource populations. We conduct a technology demonstration in simulation using the Gemini Pro model [1], showing DLM can dynamically shape policy outcomes using only human prompts as input.

## 1 Introduction

Limited resource allocation is a frequent challenge in public health settings. For instance, in the maternal and child health domain, preventative care awareness programs play a key role in reducing maternal mortality, where global rates are currently more than double the UN Sustainable Development Goal target of fewer than 70 deaths per 100K live births [2]. These essential programs help avoid preventable deaths [3, 4], yet are often operated by non-profits supporting low-resource communities [5, 6], and thereby face *two key challenges* in distributing resources to large beneficiary populations. First, programs typically operate with insufficient financial and human resources, necessitating effective resource allocation strategies to maximize health outcomes [7, 8]. Second, these programs must frequently adapt to changing population needs and intervention priorities [3]. Adaptability is especially important for prioritizing care for known subpopulations with higher risk [9], and for adjusting strategies using community or expert-driven insights [10].

To address the first problem, restless multi-armed bandits (RMAB) have proven effective in real-world deployment for optimally allocating limited resources in critical public health domains [11, 12, 13, 14]. In the classical RMAB formulation, a central planner chooses arms to allocate resources, observing a per-arm state-dependent reward. However, while RMABs are well-suited for resource allocation, a significant challenge persists in developing models capable of adapting to changing policy objectives. For instance: public health experts must often dynamically allocate resources to

---

*Equal contribution.

specific underprivileged socioeconomic groups [15, 16, 17], yet existing works in RMABs largely focus on fixed objectives, requiring significant human design to achieve new desired health outcomes.

Recently, large language models (LLMs) have emerged as adept planners in tasks such as navigation [18], spatio-temporal reasoning [19] and interactive decision-making [20]. Recent works have also demonstrated that LLMs, from language prompts, can generate reward functions as code, automating complex robotic manipulation tasks in reinforcement learning (RL) [21]. While there is growing research in LLMs for healthcare [22], the potential of LLMs to dynamically adapt resource allocation using language prompts—potentially enabling automated policy tuning using expert and community health insights [23]—remains unstudied.

In this work, we propose a Decision-Language Model (DLM) for RMABs, enabling dynamic fine-tuning of resource allocation policies for public health using human language prompts. We propose: 1) using LLMs to disambiguate language-expressed policy preferences, 2) using LLMs to directly propose reward functions as code to reach desired RMAB objectives, and 3) a fully automated self-reflection stage to iteratively refine LLM-generated reward functions using grounded RMAB simulations, without requiring ground truth feedback. Stepping beyond existing work in LLMs for health, we uniquely propose using LLMs to tune *RMAB-driven resource allocation* through reward function design, enabling: 1) more principled resource allocation using RMABs as a central planner, rather than direct LLM output, 2) continual alignment of reward functions to specified prompts using simulated RMAB allocation outcomes to refine LLM proposals, and 3) improved interpretability of proposed rewards through the use of code for reward function output.

We assess our proposed method within a *simulated* public health environment created in partnership with ARMMAN, an India-based non-profit that spreads preventative care awareness to pregnant women and new mothers through an automated call service. To increase program listenership, ARM-MAN employs health workers to provide live service calls to beneficiaries; however, due to limited support staff, ARMMAN faces a resource allocation problem to improve program listenership through optimal assignment of health workers. Prior works modeling the ARMMAN setting with RMABs have shown that RMAB policies can reduce engagement drops by $\sim 30\%$ in real-world deployment [24, 25, 26]. However, these prior works fail to address a persistent challenge within ARMMAN: tuning policies dynamically to prioritize disadvantaged groups, a common issue when targeting new regions or shifting population dynamics [27, 28]. We utilize an *anonymized* (Appendix B) dataset from ARMMAN to develop a *simulation* (Appendix C) representing changing priorities within the maternal health setting, and evaluate our method within this simulated public health environment. We provide a detailed discussion of our consideration of ethical guidelines during the design process of this work in Appendix A, B, C.

**In summary, our key contributions are as follows:**

- To the best of our knowledge, we are the first to propose using LLMs to adapt to changing resource allocation objectives in public health through reward design in the RMAB setting.
- We introduce a reward proposal loop that enhances LLM-generated reward functions using feedback from restless multi-armed bandit (RMAB) simulations, enabling LLMs to iteratively refine reward design to achieve specific, human-specified policy outcomes.
- To assess the feasibility of our system in simulation, we evaluate our algorithms' performance using the Gemini Pro model [1] in a real-world inspired task of resource allocation in maternal and child care, demonstrating near human-level policy tuning to achieve human-specified outcomes using only language prompts as input.

## 2  Related Work

**RMABs**: The RMAB problem, introduced by Whittle [29], is classically solved through the Whittle index heuristic policy [30, 31]. Subsequent works have generalized to multi-action RMABs [32, 33]. RMABs gained prominence in public health domains and are deployed to disseminate preventative healthcare information, monitor health program adherence, and model disease spread [34, 24, 35]. However, existing works focus on fixed reward functions and fail to consider that public health planners often have evolving priorities [3].

**Reward Design**: Designing reward functions that effectively condense long-term agent goals into immediate behavioral signals is a central problem in RL [36]. Manual designs are prone to task misspecification and overfitting [37]. Brys *et al.* [38] and Hussein *et al.* [39] reshape reward

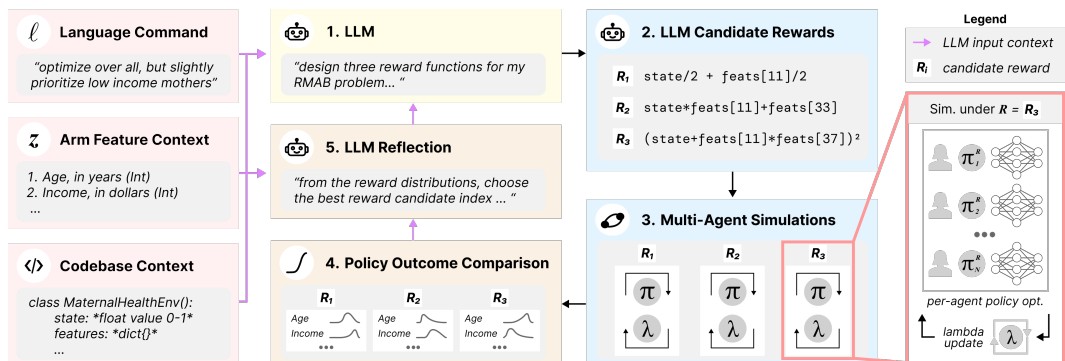

Figure 1: Overview of the DLM language-conditioned reward design loop. We provide three context descriptions to the LLM: a language command (full list of commands in Table 3), a list of per-arm demographic features available for proposed reward functions, and syntax cues enabling LLM reward function output directly in code. From this context, the *1) LLM* then proposes *2) candidate reward functions* which are used to train *3) optimal policies* under proposed rewards. Trained policies are simulated to generate *4) policy outcome comparisons* showing state-feature distributions over key demographic groups. Finally, we query an LLM to perform *5) self-reflection* [43, 21] by choosing the best candidate reward aligning with the original language command; selected candidates are used as context to guide future reward generation.

through expert behavior observation; however, expert examples may not be available in resource and data-limited public health domains. Using LLMs as dense reward signals has been investigated [40, 41]. Ma *et al.* [21] and Li *et al.* [42] further use LLMs to output reward function code from language-specified goals. However, this work fails to address multi-agent settings, where a central planner could prioritize certain groups through reward design, and where the ground truth return signal is unknown.

## 3 Background

We consider an RMAB problem with $N$ arms, each representing public health program beneficiaries, following prior work in the ARMMAN setting [24, 33, 14]. Each arm $n \in [N]$ follows a Markov decision process $(\mathcal{S}_n, \mathcal{A}_n, \mathcal{C}_n, T_n, R_n, \beta_n)$ with state space $\mathcal{S}_n$, action space $\mathcal{A}_n$, action costs $\mathcal{C}_n : \mathcal{A}_n \to \mathbb{R}$, and unknown transition probabilities $T_n : \mathcal{S}_n \times \mathcal{A}_n \to \Delta(\mathcal{S}_n)$. We define reward function $R_n : \mathcal{S}_n \to \mathbb{R}$ and discount factor $\beta \in [0, 1)$.

When $\mathcal{S}_n, \mathcal{A}_n$, and $\mathcal{C}_n$ are the same for all arms $n \in [N]$, such as the public health setting we focus on, we omit the subscript $n$. Note our algorithms still apply to more general settings. A planner learns a policy $\pi$ that maps states $\boldsymbol{s} \in S$ to actions in the set $\mathcal{A}$, given the constraint that the sum cost of actions does not exceed budget $B$. The objective is to learn a policy that maximizes the Bellman equation: $\max_{\pi \in \Pi} \left\{ J(\boldsymbol{s}) := \sum_{n=1}^{N} R(\boldsymbol{s}_n) + \beta \mathbb{E}_{s' \sim T(\cdot|s, \pi(s))} [J(\boldsymbol{s}')] \right\}$ s.t. $\pi(s_n) \in \mathcal{A}, \forall n$ and $\sum_{n=1}^{N} c_{\pi(s_n)} \leq B$ for cost of action $c_{\pi(s_n)} \in \mathcal{C}_n$. We take the Lagrangian relaxation [25], fixing an action charge $\lambda$ to decouple the value function across arms:

$$J(s, \lambda^\star) = \min_{\lambda \geq 0} \left( \frac{\lambda B}{1 - \beta} + \sum_{n=1}^{N} \max_{a_n \in \mathcal{A}} \{Q_n(\boldsymbol{s}_n, a_n, \lambda)\} \right), \tag{1}$$
$$\text{s.t. } Q_n(\boldsymbol{s}_n, a_n, \lambda) = R(\boldsymbol{s}_n) - \lambda c_{a_n} + \beta \mathbb{E}[Q_n(\boldsymbol{s}'_n, a_n, \lambda) \mid \pi(\lambda)].$$

Defined for Q-function $Q$, arm $n$, action $a_n$, transitioning from state $s_n$ under action $a_n$ to $s'_n$, and $\pi(\lambda)$ an optimal policy under $\lambda$. [43, 21]. We additionally build on deep RL approaches for RMABs [44, 33], which we discuss in Appendix F.

## 4 Decision-Language Model for RMABs

Below, we provide an overview of DLM (see Figure 1). We first describe the reward generation setting and the context provided to the LLM for initial reward candidate generation (Figure 1 Steps 1,2). We

next discuss the independent simulation stage (Step 3) and subsequent policy outcome comparison stage that simulates each policy reward outcomes (Step 4). We then discuss LLM reflection, which identifies top candidates to guide future in-context learning for LLM reward generation (Step 5). Finally, we motivate the use of LLMs for reward generation from a theoretical perspective.

## 4.1 Problem Setting: Reward Generation for RMABs

We first define the goal of language-conditioned reward design in the RMAB setting. We consider a human-specified language policy goal $\ell$ with a corresponding base reward function that maps policies $\pi \in \Pi$ to real, scalar values: $F : \Pi \to \mathbb{R}$. This base reward function $F$ establishes a scalar "ground truth" evaluation that corresponds directly to the human-specified command. Then, our key objective is, given a human-specified language prompt $\ell$ and arm features $\mathbf{z}$, to generate the reward function $R$ that maximizes base function $F$ through optimal policy $\pi^{R*}$ for reward $R$:

$$\max_{R \in \mathcal{R}} F(\pi^{R*}), \text{ where: } \pi^{R*} = \operatorname*{argmax}_{\pi \in \Pi} \left\{ \sum_{n=1}^{N} R(\boldsymbol{s}_n, \mathbf{z}) + \beta \, \mathbb{E}\left[ J\left(\boldsymbol{s}'\right) \mid \boldsymbol{s}, \pi \right] \right\}$$

$$\text{s.t. } \pi(s_n) \in \mathcal{A}, \ \forall n \quad \text{and} \quad \sum_{n=1}^{N} c_{\pi(s_n)} \leq B.$$

Note that we do *not assume access* to the ground truth reward function $F$ during training; however, we may query this ground truth "Base" reward at test time to evaluate a proposed policy. We propose using LLMs to: 1) propose reward function $R$, such that optimal policy $\pi^{R*}$, when evaluated via the unknown base function $F$, maximizes reward output, and 2) automatically refine $R$ through self-reflection.

## 4.2 Provided DLM Context

In the reward function generation phase, we prompt the LLM with three key contextual components (shown in Alg. 1, line 1). First, we include a human-language command $\ell$ that describes a desired policy outcome targeting a feature-based group (see Table 3 for examples). In practice, this is the *only human input required* for the proposed DLM technique. Second, we provide arm feature information, denoted $\mathbf{z}$. In the public health setting, features may include demographic information, socioeconomic data, or other descriptors relevant to policy outcomes; we provide the features used in our simulated setting in Appendix Figure 4. We propose that LLMs, given a language prompt $\ell$, may be used to extract the relevant features $\mathbf{z}_\ell \subseteq \mathbf{z}$ from $\ell$ to design a reward function $R$. Third, we provide context regarding the RMAB setting and code implementation. Specifically, we include information describing state, a per-arm binary value, and syntax for accessing feature information (example prompt shown in Appendix I). Using this context, we then query an LLM to propose reward functions as code (following [21, 42]) to achieve language-specified outcomes; to improve the alignment of proposed functions to these language-specified goals, we then introduce a key multi-agent simulation stage within the DLM loop.

---

**Algorithm 1** DLM

---

1:  **Input:** Task $\ell$, feature $\mathbf{z}$, and code $\zeta$ context, **Output**: $R_{\text{DLM}}$
2:  **Hyperparams:** Loop iters. $I$, proposal batch $K$, sim. epochs $n_\text{e}$, sim. steps $n_\text{s}$, num. arms $N$
3:  **for** iteration $= 1$ to $I$ **do**
4:      ## *LLM reward proposal:* Sample LLM reward funcs: $R_{1:K} \sim \text{LLM}(\ell, \mathbf{z}, \zeta)$, init. reflection string $\mathcal{Y}$
5:      ## *Multi-agent simulations:* Init. policy and critic net. $\theta, \Phi$
6:      **for** reward $i = 1$ to $K$ **do**
7:          Init. $\lambda$-net. $\Lambda$, buffer $\mathcal{D}$, states $\mathbf{s} = \mathbf{s}_0$, features $\mathbf{z}$ and clear reflection string $\mathcal{Y} = []$
8:          **for** epoch $= 1$ to $n_\text{e}$ **do**
9:              Get lambda for current states: $\lambda = \Lambda(\mathbf{s})$
10:             **for** timestep $t = 1$ to $n_\text{s}$ **do**
11:                 Sample actions: $a_n \sim \theta(s_n, \lambda) \quad \forall n \in [N]$, simulate: $\mathbf{s}', \mathbf{r} = \text{Simulate}(\mathbf{s}, \mathbf{a}, R_i, \mathbf{z})$,
12:                 update buffer $\mathcal{D} \leftarrow \mathcal{D} \cup \{(\mathbf{s}, \mathbf{a}, \mathbf{r}, \mathbf{s}', \lambda)\}$, update states: $\mathbf{s} \leftarrow \mathbf{s}'$
13:             Update $(\theta, \Phi)$ with PPO using $\mathcal{D}$ and update $\Lambda$ with $\mathcal{D}$
14:         ## *Outcome comparison (Alg. 2)*: Update $\mathcal{Y} \leftarrow \mathcal{Y} \cup OutAnalysis(\theta, \Phi, \lambda, \mathbf{Z})$
15:     ## *Top candidate selection*: $R_{\text{DLM}} \leftarrow \text{LLM}(\ell, \mathcal{Y}, \text{"choose best..."})$, $\zeta \leftarrow \zeta \cup \{R_{\text{DLM}}\}$

---

### 4.3 Multi-Agent Simulation

We evaluate each LLM-proposed reward function $R_{1:K}$ by training a policy network $\theta$ under each proposed reward $R_i$ (shown Alg. 1, lines 5:13). We consider a simulation space which defines the number of arms $N$, each with fixed, but hidden, transition dynamics. Following the procedure defined in Alg. 1, for a given reward $R_i$, we first sample an action-charge $\lambda$, which is used to decouple the learning of arm policies (Eq. 1, Alg. 1 line 9). Then, we simulate $n_s$ timesteps, sampling trajectories for each arm under the action-charge $\lambda$ (Alg. 1 lines 10:12), following [25, 45]. These trajectories are used in a buffer $\mathcal{D}$ (Alg. 1 line 12) to eventually update the policy and critic network$(\theta, \Phi)$ (Alg. 1 line 13). Note that we compute PPO advantage estimates for the actor from estimated Q-values.

### 4.4 Reflection Stage

We propose enabling LLM self-reflection using a policy outcome comparison procedure described in Appendix D Algorithm 2. Unlike prior works [21], we do not assume access to a numerical, scalar fitness function $F$ to validate reward functions. In the absence of this scalar feedback, we propose showing state-feature distributions, described below, for LLMs to select top candidate reward functions, and use these top candidates to guide future generations. Given trained policy and critic network $\theta, \Phi$, learnt lambda value $\lambda$, and feature matrix $\mathbf{Z} \in \mathbb{R}^{N \times m}$, we first simulate over $n_s$ evaluation timesteps (Alg. 2 lines 5:7). For stored accumulated agent states $\mathbf{S}$, we designate preferred "positive states"; this may be customized depending on the setting, but we consider a binary per-agent state with preferred "positive" state 1. Using accumulated states, we then compute the percentage of positive *states*, out of all positive states observed, accrued from each key feature group $g \in \mathbf{G}$ over the evaluation timesteps, as a signal for intervention effect (Alg. 2 lines 9:11). These *state-feature distributions* quantify the distribution of accumulated positive *states* over key demographic *features*, ultimately describing the percentage of total positive intervention effects attributed to each discretized age range, education level, or income bracket, for example. Each distribution is reported as an output string (see Appendix M for examples).

We then query an LLM to select the best candidate example given the original prompt $\ell$ and the outcome distribution string $\mathcal{Y}$ (Alg. 1 line 15). This selected example is then added as context in the prompt of the next iteration of reward generation (Alg. 1 line 15). Intuitively, because we focus on resource allocation in the public health domain, *we propose that simulated state-feature distributions provide sufficient signal for the LLM to analyze proposed rewards and select top candidate examples* for future in-context learning. This approach has two key strengths. First, we enable greater transparency through state-feature distribution analysis. By providing *only* simulated outcomes of policies under specified rewards $\pi^{R*}$ for reflection, we ensure that self-reflection focuses purely on alignment between the stated language goal and the health resource allocation outcomes, *without assuming any access to ground truth reward*. Second, we allow for greater flexibility; by investigating only the outcome distributions of the proposed rewards, rather than original policies themselves, we enable a model-agnostic approach to self-iteration in reward design.

### 4.5 LLM Reward Generation Capability

We theoretically justify our method and provide further insight into how LLMs generate a reward function via reflection. We propose that the LLM can be interpreted as following a hyperparameter optimization algorithm: (1) Find the relevant features from $z$ using world knowledge (2) Generate a reward function which is a weighted sum of the relevant features and observe the outcome of the optimal policy corresponding to this reward via state-feature distribution feedback (see Section 4.4) (3) Find the state-feature distribution that best conforms to the human language command (4) Optimize the weights to obtain a better candidate reward function.

We find empirically in Table 1 and Appendix H that the LLM indeed implements the first step of the proposed algorithm, by analyzing the precision and recall of finding relevant features through the usage of world knowledge and logical reasoning. This finding corroborates the results found in the logical reasoning analysis of language models done by Clark et al. [46]. We assume that the LLM can evaluate reward functions by assigning a value to their state-feature distribution. In this section, we prove by construction that a transformer can implement the second step of this algorithm and give complexity bounds and correctness guarantees of this optimization process.

In our setting, we consider the embedding space $\mathbb{R}^d$, whose elements encode states and features of an arm. We assume that there is an embedding function $\phi$ which maps features of an arm (denoted by $z$) and its state $s$ to a vector in $\mathbb{R}^d$. That is $\exists \phi : \phi(s, z) \in \mathbb{R}^d$. We consider each entry of the vector $\phi(s, z)$

to correspond to some simple single feature of the arm such as (age group) or a complex combination of simple features such as (age group, education level). Let $\mathbb{1}()$ denote the indicator function. For example, the first entry of the vector $\phi$ could be $\phi_1(s, z) = \mathbb{1}(20 \le \text{Age}(z) \le 25, s = 0)$ allowing us to set the rewards to arms in state 0 and age between 20 and 25. The 10-th entry could be $\phi_{10}(s, z) = \mathbb{1}(20 \le \text{Age}(z) \le 25, \text{Education}(z) = \text{High School}, s = 1)$, allowing us to set the rewards for arms in state 1, ages between 20 and 25 and education level being high school.

Based on empirical evidence of the LLM generated rewards, we consider the reward function for arm with features $z$ in state $s$ is of the form $R(s, z) = w^T \phi(s, z)$ for some $d$ dimensional vector $w \in \mathbb{R}^d$ with non-negative entries. This represents a rich class of rewards. In the example above, if $w_1 = 1$ and $w_{10} = 10$ and all other entries of $w$ are zeros, the reward function is:

$$R(s, z) = \begin{cases} 1 \text{ if } 20 \le \text{Age}(z) \le 25, s = 0 \\ 10 \text{ if } 20 \le \text{Age}(z) \le 25, \text{Education}(z) = \text{High School}, s = 1 \\ 0 \text{ otherwise} \end{cases} \tag{2}$$

We let the true reward function to be $R^*(s, z) = (w^*)^T \phi(s, z)$ for some $w^* \in \mathbb{R}^n$ with non-negative entries. Assume that $w^*$ is a sparse vector - i.e, the number of non-zero entries is a constant independent of $n$. This roughly means that the reward depends on only a few features. Let $\text{Supp}(w^*)$ be the set of indices where $w^*$ has non-zero entries. Let $\|w^*\|_0 := |\text{Supp}(w^*)|$.

Finding $\text{Supp}(w^*)$ can be a challenging task, since $n$ can be very large. Based on the discussion above, in Step 1, the LLM is able to use its world knowledge and find $\text{Supp}(w^*)$. We now try to understand how the LLM approximates the corresponding entries of $w^*$ via steps 2,3 and 4. Based on empirical analysis of LLM outputs, we propose the following hyperparameter search algorithm in the log-space to model its behavior. For some $\alpha > 1$, denote the search space to be $S = \{\alpha^k : -K \le k \le K\}$ for some integer $K$. The algorithm searches for $(w_i)_{i \in \text{Supp}(w^*)} \in S^{|\text{Supp}(w^*)|}$. Here, $\alpha$ represents the base of the logspace, or level of granularity of the discretization. In practice, we found that $\alpha$ was often set to be 10 by the LLM. Let $V(w, w^*)$ denote the value assigned by the LLM to the state-feature distribution corresponding to the reward $w^T \phi(s, z)$ (as proposed in Step 3). We assume that the $V(,)$ has the following monotonicity property, which roughly states that if $w$ comes closer to $w^*$, then $V(w, w^*)$ increases.

**Definition 1** (Monotonicity). *We say that the return function $V$ is monotone if for any $u, v \in \mathbb{R}^n$, if $|u_i - w_i^*| \ge |v_i - w_i^*|$ for every $i \in [n]$, then $V(u, w^*) \le V(v, w^*)$.*

**Proposition 1.** *Assume monotonicity for $V(\cdot, w^*)$ and let $\hat{w} := \arg\max_{w \in S^{|\text{Supp}(w^*)|}} V(w, w^*)$. There exists a transformer $T$ with constant depth $D$ and width $O(\|w^*\|_0 K)$ which can find $\hat{w}$ with $O(\|w^*\|_0 K)$ samples, with oracle access to $V(\cdot, w^*)$.*

This shows that the LLM has the capability to correctly optimize reward weights and converge to a good reward function, under certain assumptions. We give a detailed proof in Appendix G.

## 5 Experimental Evaluation

### 5.1 Simulated Public Health Setting

We evaluate DLM in a *simulated setting* developed in collaboration with ARMMAN, an India-based non-profit that enrolls pregnant women (beneficiaries) from low-income communities [33, 24] into an automated messaging service sharing preventative healthcare information. ARMMAN employs health workers to provide live service calls to beneficiaries with a higher risk of program drop-out, however, the beneficiary population far outnumbers the available health workers. Currently, deployed RMABs in the ARMMAN setting model beneficiaries as arms, live health worker calls as intervention actions [48, 49, 50], and binary arm states 0 as "not engaged" and 1 as a preferred "engaged" (positive) state when beneficiaries listen to an automated message for $> 30$ seconds. The RMAB policy then identifies *individuals across the entire beneficiary population* to live call such that overall program engagement is maximized [51, 26]. We consider this the old, fixed "Default" policy goal. However, a key challenge for ARMMAN is training *dynamic* policies to support specific underrepresented groups, for example when the program is deployed in different regions [27]. We simulate these dynamic policy goals with 16 distinct prompts (Table 3) that describe demographic feature-based subpopulation priorities. We simulate this environment using a fully anonymized dataset collected by ARMMAN in January 2022, created from a service quality improvement study of 7,668 mothers; we compute transition probabilities from this dataset to simulate actions of real-world beneficiaries.

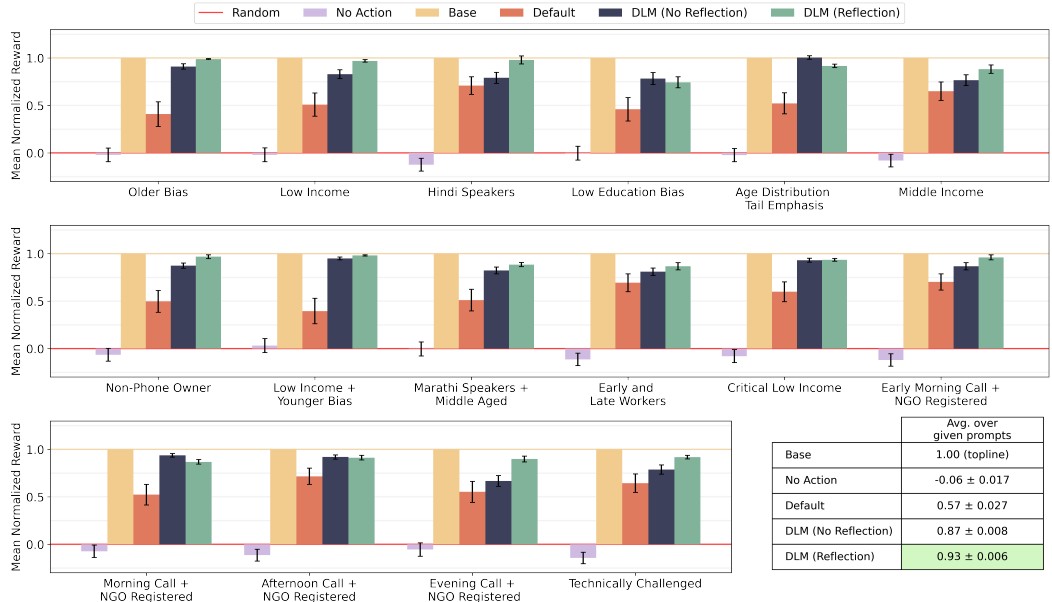

| | Avg. over given prompts |
|---|---|
| Base | 1.00 (topline) |
| No Action | -0.06 ± 0.017 |
| Default | 0.57 ± 0.027 |
| DLM (No Reflection) | 0.87 ± 0.008 |
| DLM (Reflection) | 0.93 ± 0.006 |

Figure 2: Main results. We compute normalized reward (Section 5.2) for each method over 200 seeds, and report the interquartile mean (IQM) and standard error of the IQM across all runs [47]. We compare the topline Base reward policy to the performance of DLM with No Reflection and with Reflection. We also compare to a No Action and Random policy, and a Default policy that demonstrates how the original (fixed) reward function would perform for each new task. Our method is able to achieve near-base reward performance across tasks, and consistently outperform the fixed Default reward policy in a completely automated fashion. For some tasks, DLM with Reflection is also able to significantly improve upon zero-shot proposed reward.

**Data usage and safety**: We use a fully anonymized dataset to simulate the ARMMAN public health setting; all experiments are a secondary analysis using this anonymized dataset, and are conducted with approval from the ARMMAN board of ethics. There is *no actual deployment* of this technology demonstration in the real-world ARMMAN setting. Results from a *simulated setting* are shown to highlight potential use cases of our proposed system. For more details about the dataset and consent of data collection, please see Appendix Section B and Section C.

## 5.2 Tasks, Baselines, and Metrics

We provide experimental results for 16 prompts (Table 4) which include examples emphasizing both single and multiple combined feature categories. We compute per-prompt *mean normalized reward* (described below) over 200 seeds. We evaluate DLM against several baseline reward functions:

**Random**: This policy samples arms uniformly at random, and has an MNR score of 0 through normalization. **No Action**: This policy samples no arms at each timestep, typically yielding negative MNR or near-random MNR in certain sparse reward tasks. **Default**: This policy is trained with reward prioritizing *all beneficiaries equally*, representing the old, fixed ARMMAN goal; thus, reward=1 for state 1 and reward=0 for state 0. Most prior work in RMABs assume this reward. **Base**: Reward is equal to the ground truth fitness function, a human-specified goal function described by the potentially ambiguous language prompt. At test time, we evaluate all methods using Base reward, and consider Base-trained policy as topline (MNR=1). To be clear, each method uses a different reward at train time, but uses *Base* reward at evaluation time. **DLM (No Reflection)**: This version of our proposed method ablates the reflection and iterative refinement process, allowing us to assess the importance of reflection in improving zeroshot LLM-generated reward functions. **DLM (Reflection)**: This is the full version of our proposed method.

We compute mean normalized reward (MNR) as the average reward achieved per policy, over 200 training seeds, *using base reward during test-time evaluation*. We normalize each reward $R$ as

| | Task 0: Older Bias | Task 4: Age Distribution Tail Emphasis | Task 15: Technically Challenged |
|---|---|---|---|
| **Base Reward** | 0.1**s** + 2**s** × (oldest_age) | 0.1**s** + 2**s** × (oldest_age or youngest_age) | 0.1**s** + 2**s** × (husband_phone or family_phone) |
| **DLM (No Reflection)** | 0.6**s** + 0.4**s** × (oldest_age) | 4**s²** + 10**s** × (2ⁿᵈoldest_age or oldest_age) + 17**s** × (2ⁿᵈyoungest_age or youngest_age) | 0.1**s** + 1.5**s** × (illiterate or husband_phone or evening_call) |
| **DLM (Reflection)** | **s** + 0.2**s** × (oldest_age) | 0.1**s** + 10**s** × (oldest_age) + 14**s** × (youngest_age) | 0.1**s** + 2**s** × (husband_phone and (NGO or ARMMAN or PHC affiliation)) |

Figure 3: Examples of DLM-generated reward functions vs. ground truth Base reward. Rewards reformatted for clarity; **s** represents the binary state, numbers are scalar multiplier quantities, and named features, each binary quantities, are shown. In some cases (ex: Older Bias) DLM may identify relevant features zeroshot, and use reflection to refine weights. Alternatively, reflection may help refine features (ex: Age Distribution Tail Emphasis). However, when prompts are ambiguous (ex: Technically Challenged), reflection may not have sufficient signal to effectively iterate; in these cases, additional human feedback may be required.

$(R - R_{rand})/(R_{base} - R_{rand})$. An MNR of 0 therefore implies a policy's performance equivalent to random allocation, while an MNR of 1 implies performance at the level of base reward policy.

## 5.3 Training Details

We use the Gemini Pro LLM model [1] from Google to generate reward functions, and train our downstream policy with RL using PPO (Alg. 1 line 13) [52, 45] and the generated reward function. The downstream policy is trained for $5$ epochs, with each epoch containing 100 simulation steps accumulated in a buffer. Thus, we train on a total of 500 samples for each arm. Each such training is run for each proposed reward function, over 2 rounds of reflection and 2 candidate reward functions per round. We evaluate each *prompt* over 200 seeds, with 50 separate trials per seed and 10 trajectory steps per trial, totaling $100,000$ total steps per task.

## 5.4 Evaluation of DLM Performance

We present the experimental evaluation of DLM in Figure 2. To the best of our knowledge, *we are the first to demonstrate LLMs as capable policy designers for resource allocation tasks in public health.* We make several observations on the results shown in Figure 2.

**DLM approaches Base reward performance**: We find DLM approaches Base reward performance across a broad range of tasks. Note that mean normalized reward (MNR) scores reported in Figure 2 are normalized as described in Section 5.2; this score therefore represents performance above random policy (MNR=0) compared to the topline Base reward policy (MNR=1). We find that DLM (Reflection) achieves an average MNR score of $0.93 \pm 0.006$, achieving 90% or higher of base level performance in 11/16 tasks, demonstrating to the best of our knowledge the first example of using LLMs to adapt to changing resource allocation objectives in public health through reward design.

**Reflection improves performance for many tasks**: We find that over the 16 tested prompts DLM (No Reflection) achieves an average MNR of $0.87 \pm 0.008$, while DLM (Reflection) achieves an average MNR of $0.93 \pm 0.006$. Further, 8/16 of the prompts showed a significant increase in performance after reflection (Appendix Table 5). Notably, we observe that in cases where reflection does not improve performance, DLM (No Reflection) often achieves high zero-shot reward, obviating the need for the reflection stages. For example, we observe for the *Age Distribution Tail Emphasis* prompt that DLM (No Reflection) achieves an MNR score approximately equal to the normalized Base score. Thus, DLM (Reflection) may unnecessarily iterate on functions that already achieve Base reward, resulting in degraded performance. Such occurrences are rare, and may have little practical difference in the resulting performance, which still achieves near-Base score.

**DLM consistently outperforms Default reward**: We observe that DLM consistently outperforms the Default reward policy, which represents the performance of the previously-used, fixed reward function across the given prompts. We observe an average Default reward MNR score of $0.57 \pm 0.027$, compared to $0.87 \pm 0.008$ for DLM (No Reflection) and $0.93 \pm 0.006$ for DLM (Reflection). Furthermore, we observe that in 11/16 prompts DLM (No Reflection) significantly outperforms Default policy, while in 16/16 prompts DLM (Reflection) outperforms Default (Table 5 in Appendix). In summary, we find that DLM can meaningfully adjust policies, using only language input, to align more closely to human preferences. Additionally, this highlights that the previous fixed Default

Table 1: Average precision/recall of features used in LLM-proposed reward functions compared to ground truth Base reward function. Comparison between zeroshot DLM (No Reflection) and DLM (Reflection). Cells in yellow showed improvement from Reflection with $p < 0.1$; cells in green showed improvement from Reflection with $p < 0.05$. Results indicate LLMs are very effective feature extractors for reward function generation. Furthermore, the Reflection module is particularly useful for improving recall rates, as 13/16 tasks showed significant recall improvement with Reflection.

| Task | 0 | 1 | 2 | 3 | 4 | 5 | 6 | 7 |
|---|---|---|---|---|---|---|---|---|
| Prec. (zeroshot) | $0.49 \pm 0.028$ | $0.27 \pm 0.016$ | $0.93 \pm 0.014$ | $0.45 \pm 0.022$ | $0.81 \pm 0.019$ | $0.48 \pm 0.017$ | $0.07 \pm 0.014$ | $0.33 \pm 0.017$ |
| Prec. (reflection) | $0.54 \pm 0.026$ | $0.29 \pm 0.015$ | $0.87 \pm 0.017$ | $0.45 \pm 0.022$ | $0.84 \pm 0.018$ | $0.54 \pm 0.016$ | $0.11 \pm 0.017$ | $0.38 \pm 0.015$ |
| Rec. (zeroshot) | $0.72 \pm 0.032$ | $0.64 \pm 0.034$ | $1.00 \pm 0.000$ | $0.87 \pm 0.024$ | $0.89 \pm 0.016$ | $0.74 \pm 0.025$ | $0.11 \pm 0.021$ | $0.67 \pm 0.027$ |
| Rec. (reflection) | $0.84 \pm 0.026$ | $0.72 \pm 0.032$ | $1.00 \pm 0.000$ | $0.92 \pm 0.020$ | $0.94 \pm 0.013$ | $0.84 \pm 0.020$ | $0.17 \pm 0.026$ | $0.80 \pm 0.020$ |
| Task | 8 | 9 | 10 | 11 | 12 | 13 | 14 | 15 |
| Prec. (zeroshot) | $0.78 \pm 0.018$ | $0.41 \pm 0.012$ | $0.93 \pm 0.014$ | $0.87 \pm 0.015$ | $0.93 \pm 0.012$ | $0.97 \pm 0.009$ | $0.96 \pm 0.010$ | $0.04 \pm 0.009$ |
| Prec. (reflection) | $0.83 \pm 0.015$ | $0.42 \pm 0.010$ | $0.93 \pm 0.013$ | $0.84 \pm 0.013$ | $0.95 \pm 0.011$ | $0.96 \pm 0.009$ | $0.94 \pm 0.011$ | $0.06 \pm 0.010$ |
| Rec. (zeroshot) | $0.65 \pm 0.015$ | $0.46 \pm 0.014$ | $0.43 \pm 0.013$ | $0.97 \pm 0.009$ | $0.96 \pm 0.009$ | $0.98 \pm 0.006$ | $0.98 \pm 0.006$ | $0.09 \pm 0.017$ |
| Rec. (reflection) | $0.74 \pm 0.013$ | $0.51 \pm 0.011$ | $0.54 \pm 0.015$ | $0.98 \pm 0.006$ | $0.99 \pm 0.004$ | $1.00 \pm 0.002$ | $0.99 \pm 0.005$ | $0.13 \pm 0.020$ |

reward is ineffective in accommodating the diverse prompts, and hence policy goals, described by humans in our simulated setting.

## 5.5 LLM-Generated Rewards and Reflection in Public Health Settings

In this section we provide insight into LLM-generated reward functions and the reward Reflection module. We find that LLMs can reliably design and improve upon reward functions for public health goals, enabling automated policy tuning in resource-limited settings.

**LLMs accurately interpret language prompts for policy shaping**: We provide examples of LLM-generated reward functions compared to ground truth Base rewards in Figure 3. We find that DLM can consistently capture the ground truth Base reward, and use reflection to improve upon these proposed functions. However, in cases where language is ambiguous (e.g. "Technically Challenged" example), inherent language ambiguities may result in misaligned proposals. These findings suggest that LLMs can automatically tune public health allocation policies using only language-described preferences. Ultimately, while human input may be required to overcome ambiguities of language prompts, DLM allows a human decision-maker to monitor state-feature distributions and iteratively provide expert opinion to guide LLM-proposed policies.

**LLMs as feature extractors for reward proposal in resource allocation tasks**: We find LLMs can accurately extract relevant features from language-described resource allocation preferences. We demonstrate these results in Table 1. LLM-proposed rewards have consistently high precision and recall, effectively capturing the features used in the Base reward function. We find in some cases, for instance Task 15 (Technically Challenged), that ambiguous prompts (see Appendix Table 4) may lead to lower recall ability. Interestingly, however, we still observe reasonable performance in these cases (Figure 2), suggesting that DLM may identify potentially correlated substitute features even for challenging prompts. Note, additionally, that effective feature extraction does not necessarily imply the correct usage of features; however, the findings suggest that LLMs are effective at the first critical "filtering" step of reward design, particularly in our task of public health resource allocation.

## 5.6 Limitations

We test our method in a purely simulated environment; any potential consideration of real-world deployment would require comprehensive field testing after approvals from relevant ethics boards. Additionally, we test with prompts in English; further testing in local (Indian) languages is required. Furthermore, whereas we present results on DLM with Chain-of-Thought (CoT) reasoning in Table 6, indicating no significant improvement due to CoT, we recognize that additional LLM reflection techniques have the potential to improve DLM. Finally, in any potential deployment of the proposed method, we expect that health experts should monitor state-feature distributions to intervene on proposed policies, ensuring safety and proper disambiguation of unclear input prompts.

## 5.7 Ethical Considerations in the Use of DLM

Extending our discussion of the ethical considerations we take in developing DLM (Appendix A, B, C), we further consider the broader implications of algorithmic resource allocation, particularly in public health. We note the need for mitigating data bias in the health setting, especially to

minimize harmful discrimination for underrepresented groups [53], and to avoiding data bias by enabling participatory design [54], all key considerations we make in collaboration with our partner NGO for this work. We further highlight the importance of democratized decision-making criteria for algorithmic allocation techniques [55, 56], as well as the importance of complete beneficiary autonomy through guaranteed consent and the opportunity to deny allocations [57], as we do in this work. Finally, one extension of this work that future studies may consider is to directly incorporate prior work in fairness guarantees in resource allocation settings [58, 59, 60, 61].

## 6    Conclusion

In this paper we introduce a Decision-Language Model (DLM) for resource allocation in public health, which can take language prompts describing complex, potentially ambiguous public health policy goals as input, and generate downstream control policies specialized for such goals. We uniquely demonstrate, beyond existing work in LLMs for public health, the strength of LLMs to shape *principled, RMAB-based resource allocation strategies*, potentially enabling rapid community-driven policy adjustment in critical, resource-constrained public health settings. For all future work considering potential deployment, DLM enables health experts to monitor state-feature distributions to intervene on proposed policies and guide LLM generation to ensure safety.

## Acknowledgments and Disclosure of Funding

This work was supported by the Harvard Data Science Initiative.

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

# Appendix

## A  Social Impact Statement

The presented methods carries a positive societal impact, allowing non-profit organizations in the public health domain to efficiently allocate limited health resources to large beneficiary populations, and to effectively adapt policy outcomes. We illustrate the application of our method in collaboration with an India-based public health organization promoting preventive care for pregnant mothers, that currently relies on RMAB policies for ensuring spread of important health information to low resource populations.

Our proposed methods, tested purely as a technology demonstration and entirely in simulation, do not have direct negative societal implications. However, reinforcement learning agent training should be done responsibly, especially given the safety concerns associated with agents engaging in unsafe or uninformed exploration strategies. While the public health domain we considered does not have concerns associated with extreme environments, when adapting our proposed methods for other domains, ensuring a robust approach to training reinforcement models is critical. All work in simulation was conducted in collaboration with ARMMAN's team of ethics, and any future consideration of deployment would require a real-world evaluation together with the domain partner, which may reveal further challenges to be addressed.

## B  ARMMAN Dataset Description

To conduct the secondary analyses presented in this study, we use a real dataset collected by ARMMAN (2022). This dataset was created from a comprehensive quality improvement study carried out by ARMMAN in 2022; the study involved 7,668 beneficiaries who were enrolled over a study period of 7 weeks. Throughout this period, 20% of participants were allocated at least one direct interaction, such as a live service call from a healthcare volunteer. Engagement during this period was used to compute transition dynamics.

### B.1  Associated Features and Protected Information

Each beneficiary enrolled in the ARMMAN study was described by 43 features, which include data on age, income, education, preferred call times, and ownership of a phone, among other factors. These features are entirely anonymized. ARMMAN, through the mobile voice call service initiative, works specifically with socially disadvantaged populations. However, *ARMMAN does not collect constitutionally protected and sensitive categories such as cast and religion* in the recorded beneficiary features. In addition to such categories not being available in the ARMMAN dataset, we further ignore already-anonymized information regarding pregnancy history in our study to focus primarily on broader socio-economic categorizations such as age and income. We additionally worked with ARMMAN to ensure that the evaluated prompts, tested only in simulation, are closely aligned with health expert goals and challenging real-world scenarios where prioritizing specific underrepresented subpopulations is desired.

## C  Dataset Consent for Collection and Analysis

We provide further information regarding consent for data collection and analysis below.

### C.1  Secondary Analysis

In this study, we conduct a *secondary analysis* of the ARMMAN dataset. We use trajectories, which contain the historical engagement behavior for beneficiaries enrolled in the ARMMAN study. Engagement states are computed as binary values, which are determined as "engaged" if a beneficiary listens to a voice message for more than 30 seconds. The average length of each voice message is 115 seconds. For each beneficiary, we utilize *only an estimated transition dynamic matrix*, computed using the historical trajectories for $T = 7$ weeks. We only interface with this estimated transition matrix per beneficiary; for each sampled arm then, we observe only the provided transition matrix and the associated, anonymized, arm features. We then use these estimated transition matrices to

*simulate* agent behavior; we *do not deploy the proposed method to the ARMMAN program.* These described secondary experiments are conducted completely in simulation using estimated transition matrices to simulate plausible real-world agents.

## C.2 Data Collection Consent

Full consent for data collection is obtained from each ARMMAN study participant; the process of data collection is explained in detail to each participant when obtaining consent, *prior to data collection.* The dataset was completely anonymized by ARMMAN, and data exchange from ARMMAN to the researchers was regulated through clearly defined exchange protocols including anonymization, read-only researcher access, restricted use of data for research purposes only, and approval by the ARMMAN ethics review committee.

## C.3 Simulated Findings and Equal Distribution of Resources

We note again that all experiments presented in this work were conducted in simulation; we attempt to simulate a plausible real-world public health setting to gain insight into strengths and weaknesses of the proposed approach. We additionally note that the intended goal of this research is to identify techniques to more easily align algorithmic resource allocation methods to specific goals of public health experts. However, the service call program is always equally accessible to all beneficiaries (e.g., participants can always request service calls via a free missed call service) and health information disseminated through the program will never be restricted for any individual enrollee. *That is, all participants will receive the same weekly health information by automated message regardless of whether they are scheduled to receive service calls or not.* The proposed system is intended for use *only in cases where separate, additional service call resources are available* to help especially underrepresented groups. In these cases, the proposed system may help align algorithmic resource allocation of the *separate* additional resources more closely to desired public health expert policy goals.

# D Outcome Analysis Algorithm

Below, we provide the algorithm for the state-feature distribution analysis used in the DLM reflection stage to select top candidate rewards. Please see subsection 4.4 for more details on the Reflection module and state-feature analysis procedure.

---

**Algorithm 2** OUTCOMEANALYSIS

---

1: **Input:** Trained policy, critic net. $\theta, \Phi$, action-charge $\lambda$, feature matrix $\mathbf{Z} \in \mathbb{R}^{N \times m}$, **Return:** OutString
2: **Hyperparameters:** Sim. steps $n_s$
3:   ## *Step 1: Simulate $n_s$ timesteps under $\theta$*
4: Init. evaluation totals $\mathbf{S} \leftarrow [0] \times N$, percentage distribution $\mathbf{P} \in \mathbb{R}^{|\mathbf{G}|}$, OutString as an empty string
5: **for** timestep $t = 1$ to $n_s$ **do**
6:     Sample actions: $a_n \sim \theta(s_n, \lambda) \quad \forall n \in [N]$, simulate actions: $\mathbf{s}', \mathbf{r} = \text{Simulate}(\mathbf{s}, \mathbf{a}, R_i, \mathbf{z})$,
7:     accumulate states: $\mathbf{S} \leftarrow \mathbf{S} + \mathbf{s}$, update states: $\mathbf{s} \leftarrow \mathbf{s}'$
8:   ## *Step 2: Compute state outcome distributions*
9: $\mathbf{G} \leftarrow$ Define feature groups within $\mathbf{Z}$
10: **for** each group $g \in \mathbf{G}$ **do**
11:     Compute state distribution over feats: $\mathbf{P}(g) \leftarrow \frac{\sum_{i \in g} \mathbf{S}_i}{\sum \mathbf{S}}$, append to OutString: "$g$: $\mathbf{P}(g)$% of reward; "

---

# E Compute Resources

We use the Gemini Pro LLM model [1] from Google to generate reward functions. We use 8 CPUs cores (Intel Cascade Lake Processors) for all tasks.

# F    Deep RL for RMABs

The Lagrangian relaxation allows us to disentangle the learning of arm policies and an action charge $\lambda$. To learn arm policies, it is common to use RL approaches such as tabular Q-learning, deep Q-learning, or policy gradient algorithms[44]. In this paper, we use PPO [52], a policy gradient algorithm with an actor-critic structure. Specifically, the Q-function can be updated as follows:

$$
\begin{aligned}
Q_n^t(s_n, a_n) \leftarrow &(1 - \alpha_q)Q_n^{t-1}(s_n, a_n) \\
&+ \alpha_q \left( R(s_n) - \lambda c_{a_n}, \beta \max_a Q(s_n', a) \right),
\end{aligned}
$$

where $\alpha_q$ is the learning rate. For an actor-critic methods, after performing an update of the Q-function in the critic, one may compute the advantage estimate:

$$
A\left(\mathbf{s}, \mathbf{a}\right) = Q\left(\mathbf{s}, \mathbf{a}\right) - V\left(\mathbf{s}\right).
$$

In practice, the values $A(s, a), Q(s, a), V(s)$ will be evaluated based on the current policy $\pi$. Intuitively, the advantage estimate tells us how much better $\mathbf{a}$ is compared to the current policy $\pi$. The advantage estimate is then plugged into the policy gradient to update the actor. To learn an action charge $\lambda$, we follow an updating rule [33]:

$$
\Lambda_t \leftarrow \Lambda_{t-1} - \alpha_\Lambda \left( \frac{B}{1 - \beta} + \sum_{n=1}^N \sum_{t=0}^H \mathbb{E}\beta^t c_{n,t} \right),
$$

where $c_{n,t}$ is the cost of the action taken by the optimal policy on arm $n$ in round $t$, and $\alpha_\Lambda$ is the step size.

# G    Proof of Proposition 1

Here we give a proof of Proposition 1.

**Proof sketch**. To prove this, we first show that there exists parameters $\theta$ which can implement the optimization step of a simple and efficient gridsearch algorithm ITERATEDLINESEARCH in the discretized weight space. We can do this by leveraging previous work connecting transformers and traditional algorithms [62, 63, 64]. Specifically, the RASP [64] programming language and TRACR compiler [62] enables compiling a program into transformer weights, giving a direct proof of existence by construction. We then prove the correctness of this algorithm using the monotonicty property. Finally, we prove upper bounds on the runtime of this algorithm, yielding the desired sample complexity bound in Proposition 1.

First we state our gridsearch algorithm ITERATEDLINESEARCH. In our notation, $w_i$ is the entry of $w \in \mathbb{R}^n$ corresponding to index $i \in [n]$. Recall that there is a total of $|\mathsf{Supp}(w^*)|$ weights corresponding to the non-zero entries of $w^*$, and that we are searching over $2K$ possible values for each weight. This means in total there are $(2K)^{|\mathsf{Supp}(w^*)|}$ total possible weight combinations.

---

**Algorithm 3** ITERATEDLINESEARCH

---

1: **Input:** Discretized grid of possible weights $S$, discretization level $\alpha$, evaluation function $V(w) :=$ $V(\cdot, w^*)$, support $\mathsf{Supp}(w^*)$.
2: **Return:** Optimized weights $w$ maximizing $V(w)$.
3: ## *Step 1: Set w to min value*
4: $w_i, w_i' \leftarrow \min S \quad \forall i \in \mathsf{Supp}(w^*)$
5: i = 0
6: converged = **False**
7: **while** not converged **do**
8:    ## *Step 2: Call external oracle V(w)*
9:    $V_w, V_{w'}, \leftarrow V(w, w^*), V(w', w^*)$
10:    $w_i, w_i',$ converged, $i \leftarrow$ OPTIMIZESTEP$(S, w, w', V_w, V_w', \alpha, i)$
11: ## *Step 3: Return w as output*
12: $w^{\mathsf{out}} \leftarrow w$
13: **return** $w^{\mathsf{out}}$

---

We now state the actual optimization subroutine of ITERATEDLINESEARCH, OPTIMIZESTEP.

---

**Algorithm 4** OPTIMIZESTEP

---

1: **Input:** $S^{|\mathsf{Supp}(w_0^*)|}, w, w', V(w, w^*), V(w', w^*), \alpha, i$
2: **Return:** Updated weights $w, w'$, where $V(w', w^*) > V(w, w^*)$, convergence status, and current index $i$.
3: **if** $v(w') > v(w)$ **and** $w_i < \max S$ **then**
4:    $w \leftarrow w'$   ## *update w to w' if it has a higher value*
5:    $w_i' \leftarrow w_i \alpha$
6:    **return** $w, w',$ **False**, $i$   ## *return w, w' and converged = False, i*
7: **else**
8:    $i \leftarrow i + 1$
9:    **if** $i > \|w^*\|_0$ **then**
10:      **return** $w, w',$ **True**, $i$   ## *return w, w' and converged = True, i*
11:    **else**
12:      **return** $w, w',$ **False**, $i$   ## *return w, w' and converged = False, i*

---

**Proposition 2.** *Given oracle access to a valuation function $V(w, w^*)$, OPTIMIZESTEP can be implemented in a transformer with constant depth and $O(\|w^*\|_0 K)$ width.*

*Proof.* Using the RASP programming language [64], we show that OPTIMIZESTEP can be implemented in a transformer. In each forward pass of the transformer, we take one new weight combination and pass this to the loss function in order to evaluate its performance on line 8 of Algorithm 4. Note that the only state needed to be kept track of between forward passes of the transformer is the current hyperparameter dimension $i$, $\alpha$, $S^{|\mathsf{Supp}(w_0^*)|}, w, w', V(w, w^*), V(w', w^*)$. This information can be stored in $nK$ hidden dimensions, as the size of $S^{|\mathsf{Supp}(w_0^*)|}$ upper bounds everything else as $nK$. The conditional statements on line 3 and line 9 of Algorithm 4can be implemented in two select-aggregate blocks [64], corresponding to two attention layers. Line 8 requires an additional fully connected layer after the first attention layer, and a finally an output layer is needed to implement each return statement, giving us four layers in total. The select block roughly corresponds to the attention matrix in the transformer, and is a function that allows calculating conditionals by taking in three arguments: keys, queries, and a boolean predicate $p$. Here, we would pass $\alpha$, $S^{|\mathsf{Supp}(w_0^*)|}, w,$ $w', V(w, w^*), V(w', w^*)$ as the input sequence, and construct three selectors corresponding to the boolean predicates on line 5, 6, and 8, respectively. An aggregate can then be performed to get the result of the predicate, controlling the program flow for selecting. $\square$

**Proposition 3.** ITERATEDLINESEARCH *converges in $O(\|w^*\|_0 K)$ time.*

*Proof.* Inside each iteration of the while loop, line 10 either $i$ increases to $i+1$ (but never decreases) or $w_i'$ increases to $w_i'\alpha$. There are $2K$ possible values for $w_i'$ for every given $i$. There are $\|w^*\|_0$ possible

values for $i$ and $2K$ possible values for $w'$, ensuring that the algorithm converges in $O(\|w^*\|_0 K)$ time. $\qquad\square$

*Proof of Proposition 1.* We consider iterated line search (Algorithm 3) above with search space $S$ and scale $\alpha$. In Proposition 3, we have shown that this algorithm converges in time $O(nK)$ and that this algorithm can be implemented by a transformer. It remains to show that it returns $\hat{w} := \arg\max_{w \in S^{|\mathsf{Supp}(w^*)|}} V(w, w^*)$.

By monotonicity, it is clear that if the final output is such that $w_i^{\mathsf{out}} = \arg\min_{g \in S} |g - (w^*)_i|$ for every $i \in \mathsf{Supp}(w^*)$, then $w^{\mathsf{out}} = \hat{w}$. Therefore, we prove the coordinate-wise optimality below.

Consider the value of $w$ in the while loop, line 10 when $i$ is increased to $i+1$ by the OPTIMIZESTEP function.

**Case 1:** Suppose the coordinate $w_i = \max S$ then we have $V(w, w^*) \geq V(\bar{w}, w^*)$ for every $\bar{w}$ such that $\bar{w}_j = w_j$ for $i \neq j$ and $\bar{w}_i \in S$. Therefore, by the contrapositive of the monotonicity property, this implies $w_i = \arg\min_{g \in S} |g - (w^*)_i|$. This establishes optimality for the co-orindate $i$ since $w_i^{\mathsf{out}} = w_i$.

**Case 2:** Now suppose that $w_i < \max S$. A similar argument as above establishes that $|w_i - w_i^*| \leq |g - w_i^*|$ for every $g \in S$ such that $g \leq w^i$. The loop breaks since the condition in line 8 is false. Again, by the contrapositive of the monotonicity property, we must have $|w_i - w_i^*| \leq |\alpha w_i - w_i^*|$. This implies that $w_i^* < \alpha w_i$ and hence $|w_i - w_i^*| \leq |\alpha w_i - w_i^*| \leq |\alpha^h w_i - w_i^*|$ for all $h \in \mathbb{N}$. Thus, we demonstrate that even in this case $w_i = \arg\min_{g \in S} |g - w_i^*|$.

Therefore, coordinate-wise optimality holds for every $i \in \mathsf{Supp}(w^*)$, completing the proof of the claim. $\qquad\square$

## H   Recall of Logical Combinations

We analyze the logical reasoning ability of the LLM to combine atomic features together in order to recover the ground-truth reward in Table 2.

Table 2: Recall of logical combinations of features for multi-feature prompts. We consider multi-feature prompts 4-15, and report the recall compared to Base reward for accurately emulating behavior of Base reward. Note that we consider only LLM generations with high feature recall, i.e. those proposed rewards that include, at minimum, the features used in the corresponding ground truth Base reward.

| Task | 4 | 5 | 6 | 7 | 8 | 9 |
|---|---|---|---|---|---|---|
| **Logic Recall** | $0.85 \pm 0.017$ | $0.75 \pm 0.023$ | $0.69 \pm 0.037$ | $0.69 \pm 0.027$ | $0.84 \pm 0.043$ | $0.66 \pm 0.04$ |
| **Task** | 10 | 11 | 12 | 13 | 14 | 15 |
| **Logic Recall** | $0.64 \pm 0.053$ | $0.85 \pm 0.014$ | $0.87 \pm 0.013$ | $0.9 \pm 0.01$ | $0.87 \pm 0.011$ | $0.62 \pm 0.102$ |

## I   Sample Input Prompt

A sample input prompt with full context inputs is shown below.

```
Create a Python reward function for RL in phone call resource allocation to
mothers in India, with the objective of prioritizing higher states and: While
still prioritizing all, slightly focus on the oldest by age distribution.. The
function should use 'state' (value is either 0,1) and features 'agent_feats'
(length 43 array) to direct the RL agent. Here is a description of the features
you may use:
Index Name DataType
[sensitive feature hidden]
 7. Ages 10-20 - Binary
 8. Ages 21-30 - Binary
 9. Ages 31-40 - Binary
 10. Ages 41-50 - Binary
 11. Ages 51-60 - Binary
 12. Speaks Hindi - Binary
 13. Speaks Marathi - Binary
 14. Speaks Gujurati - Binary
```

```
15. Speaks Kannada - Binary
16. Education level 1/7 -- illiterate - Binary
17. Education level 2/7 -- 1-5th Grade Completed - Binary
18. Education level 3/7 -- 6-9th Grade Completed - Binary
19. Education level 4/7 -- 10th Grade Passed - Binary
20. Education level 5/7 -- 12th Grade Passed - Binary
21. Education level 6/7 -- Graduate - Binary
22. Education level 7/7 -- Post graduate - Binary
23. Phone owner 0 (e.g., woman) - Binary
24. Phone owner 1 (e.g., husband) - Binary
25. Phone owner 2 (e.g., family) - Binary
26. To be called from 8:30am-10:30am - Binary
27. To be called from 10:30am-12:30pm - Binary
28. To be called from 12:30pm-3:30pm - Binary
29. To be called from 3:30pm-5:30pm - Binary
30. To be called from 5:30pm-7:30pm - Binary
31. To be called from 7:30pm-9:30pm - Binary
32. NGO - Binary
33. ARMMAN - Binary
34. PHC - Binary
35. Income bracket -1 (no income) - Binary
36. Income bracket 1 (e.g., 0-5000) - Binary
37. Income bracket 2 (e.g., 5001-10000) - Binary
38. Income bracket 3 (e.g., 10001-15000) - Binary
39. Income bracket 4 (e.g., 15001-20000) - Binary
40. Income bracket 5 (e.g., 20001-25000) - Binary
41. Income bracket 6 (e.g., 25001-30000) - Binary
42. Income bracket 7 (e.g., 30000-999999) - Binary
Your task:
1. Write a simple, single-line Python reward function. Exclude the word 'return'
and exclude non-standard libraries. Format your code with triple $ signs:
$$$[YOUR FUNCTION]$$$.
2. Provide an explanation on how this function prioritizes the specified age
group. Format your explanation with triple % signs: %%%[YOUR EXPLANATION]%%%.
Note that HIGHER states are always preferred, so ensure reward increases as
state increases. Make sure reward is always positive and increasing with state.
Avoid using bitwise operators &, |. Using and, or instead.
Example Prompt: While prioritizing all, emphasize agents that are both older and
richer
Let's think about this step by step. We want to give reward only for agents that
are older, which corresponds to feature 11, and rich which corresponds to
feature 42. This corresponds to a condition of (agent_feats[11] and
agent_feats[42]). In addition, we always only want to give reward when the state
is 1, since the agent gets reward only when it is in a listening state.
Therefore, our reward function should be: state * (agent_feats[11] and
agent_feats[42]).
Example Response:
Python Code: '$$$ state * 0.1 + 2 * state * (agent_feats[11] and
agent_feats[42]) $$$'
Explanation: %%%This function gives higher rewards for higher states and higher
ages, aligning with the goal to reward older individuals with higher states.%%%
Come up with a unique new reward for the specified goal: While still
prioritizing all, slightly focus on the oldest by age distribution.. Here are
your best previous attempts:
```

## J   Prompts and Base Reward Functions

We provide a full list of the tested prompts and the ground truth Base reward functions in Table 3. We additionally include a full list of the features available for each arm (and used within proposed reward functions) in Figure 4.

We note that we evaluate on a wide variety of tasks within the real of resource allocation for maternal health. In particular, we test across feature characteristics such as age, income, spoken languages (an important characteristic indicating ethnicity in the Indian subcontinentuuh), education, and other relevant features that may be considered in real-world allocation tasks. We provide these tasks directly as context to the LLM, in addition to relevant feature characteristics and their corresponding descriptions. Per seed, we fix the randomness in the sampled set of 48 arms from ARMMAN's dataset, the deep RL neural network initialization, batch sampling from the dataset, and the sampling of states for each arm during evaluation. However please note that we cannot control the randomness of the Gemini LLM generation as we only have access through the Google API interface.

```
[sensitive features 0-6 hidden, not used in prompts]
 7. Ages 10-20 - Binary
 8. Ages 21-30 - Binary
 9. Ages 31-40 - Binary
10. Ages 41-50 - Binary
11. Ages 51-60 - Binary
12. Speaks Hindi - Binary
13. Speaks Marathi - Binary
14. Speaks Gujurati - Binary
15. Speaks Kannada - Binary
16. Education level 1/7 -- illiterate - Binary
17. Education level 2/7 -- 1-5th Grade Completed - Binary
18. Education level 3/7 -- 6-9th Grade Completed - Binary
19. Education level 4/7 -- 10th Grade Passed - Binary
20. Education level 5/7 -- 12th Grade Passed - Binary
21. Education level 6/7 -- Graduate - Binary
22. Education level 7/7 -- Post graduate - Binary
23. Phone owner 0 (e.g., woman) - Binary
24. Phone owner 1 (e.g., husband) - Binary
25. Phone owner 2 (e.g., family) - Binary
26. To be called from 8:30am-10:30am - Binary
27. To be called from 10:30am-12:30pm - Binary
28. To be called from 12:30pm-3:30pm - Binary
29. To be called from 3:30pm-5:30pm - Binary
30. To be called from 5:30pm-7:30pm - Binary
31. To be called from 7:30pm-9:30pm - Binary
32. NGO - Binary
33. ARMMAN - Binary
34. PHC - Binary
35. Income bracket -1 (no income) - Binary
36. Income bracket 1 (e.g., 0-5000) - Binary
37. Income bracket 2 (e.g., 5001-10000) - Binary
38. Income bracket 3 (e.g., 10001-15000) - Binary
39. Income bracket 4 (e.g., 15001-20000) - Binary
40. Income bracket 5 (e.g., 20001-25000) - Binary
41. Income bracket 6 (e.g., 25001-30000) - Binary
42. Income bracket 7 (e.g., 30000-999999) - Binary
```

Figure 4: Full list of features with corresponding feature indices used in Base and LLM-proposed reward functions. See Table 3 for a full list of language prompts and ground truth Base reward functions.

# K  Extended Results and Statistical Tests

## K.1  Numerical Results

We show the full set of tasks and numerical results for the mean normalized reward score in Table 4.

Table 3: Full task list and ground truth Base reward functions.

| idx | Label | Full Prompt | Base Reward Function |
|---|---|---|---|
| 0 | Older Bias | While still prioritizing all, slightly focus on the oldest by age distribution. | lambda feature, state: state * 0.1 + if_(state) * 2.0 * feature[11] |
| 1 | Low Income | While still prioritizing all, slightly focus on the bottom 12.5% of the income_bracket distribution. | lambda feature, state: state * 0.1 + if_(state) * 2.0 * feature[36] |
| 2 | Hindi Speakers | While still prioritizing all, slightly focus on those who speak Hindi. | lambda feature, state: state * 0.1 + if_(state) * 2.0 * if_(feature[12]) |
| 3 | Low Education Bias | While still prioritizing all, slightly weight those who have had low education. | lambda feature, state: state * 0.1 + if_(state) * 2.0 * if_(feature[16]) |
| 4 | Age Distribution Tail Emphasis | While still prioritizing all, slightly focus on both the youngest and oldest by age. | lambda feature, state: state * 0.1 + if_(state) * 2.0 * if_(feature[11] or feature[7]) |
| 5 | Middle Income | While still prioritizing all, slightly prefer the income bracket bounds for the middle 40% of the population. | lambda feature, state: state * 0.1 + if_(state) * 2.0 * if_(feature[38] or feature[39] or feature[40]) |
| 6 | Non-Phone Owner | While still prioritizing all, slightly favor those women who do not own their own phone. | lambda feature, state: state * 0.1 + if_(state) * 2.0 * if_(feature[24] or feature[25]) |
| 7 | Low Income + Younger Bias | While still prioritizing all, slightly prioritize impoverished younger mothers by combining the distributions of 'age' and 'education'. | lambda feature, state: state * 0.1 + if_(state) * 2.0 * if_(feature[7] and feature[16]) |
| 8 | Marathi Speakers + Middle Aged | While still prioritizing all, slightly focus on those Marathi-speakers with middle-aged mothers. | lambda feature, state: state * 0.1 + if_(state) * 2.0 * if_(feature[13] and (feature[9] or feature[10])) |
| 9 | Early and Late Workers | While still prioritizing all, slightly emphasize beneficiaries who likely work early in the morning and late at night. | lambda feature, state: state * 0.1 + if_(state) * 2.0 * if_(feature[26] or feature[28]) |
| 10 | Critical Low Income | While still prioritizing all, slightly weight the lowest income_bracket groups, the absolute lowest earners in the population. | lambda feature, state: state * 0.1 + if_(state) * 2.0 * if_(feature[35] or feature[36] or feature[37]) |
| 11 | Early Morning Call + NGO Registered | While still prioritizing all, slightly advantage those who prefer being called before 10:30am 'slot' and are registered at an NGO. | lambda feature, state: state * 0.1 + if_(state) * 2.0 * if_(feature[26] and feature[32]) |
| 12 | Morning Call + NGO Registered | While still prioritizing all, slightly advantage those who prefer being called between 10:30am-12:30pm and are registered at an NGO. | lambda feature, state: state * 0.1 + if_(state) * 2.0 * if_(feature[27] and feature[32]) |
| 13 | Afternoon Call + NGO Registered | While still prioritizing all, slightly advantage those who prefer being called between 12:30pm-3:30pm and are registered at an NGO. | lambda feature, state: state * 0.1 + if_(state) * 2.0 * if_(feature[28] and feature[32]) |
| 14 | Evening Call + NGO Registered | While still prioritizing all, slightly advantage those who prefer being called after 7PM 'slot' registered at an NGO. | lambda feature, state: state * 0.1 + if_(state) * 2.0 * if_(feature[31] and feature[32]), |
| 15 | Technically Challenged | While still prioritizing all, infer technical challenges in reaching the phone that could indicate 'at-risk' beneficiaries and give slight preference. | lambda feature, state: state * 0.1 + if_(state) * 2.0 * if_(feature[24] or feature[25]) |

Table 4: Full prompts and numerical results.

| Idx. | Label | Full Prompt | Base (topline) | No Action | Default | DLM (No Reflection) | DLM (Reflection) |
|---|---|---|---|---|---|---|---|
| 0 | Older Bias | While still prioritizing all, slightly focus on the oldest by age distribution. | 1.00 | -0.02 ± 0.072 | 0.41 ± 0.130 | 0.91 ± 0.029 | 0.99 ± 0.005 |
| 1 | Low Income | While still prioritizing all, slightly focus on the bottom 12.5% of the income_bracket distribution. | 1.00 | -0.02 ± 0.073 | 0.51 ± 0.122 | 0.83 ± 0.046 | 0.97 ± 0.014 |
| 2 | Hindi Speakers | While still prioritizing all, slightly focus on those who speak Hindi. | 1.00 | -0.12 ± 0.067 | 0.71 ± 0.093 | 0.79 ± 0.058 | 0.98 ± 0.042 |
| 3 | Low Education Bias | While still prioritizing all, slightly weight those who have had low education. | 1.00 | -0.00 ± 0.073 | 0.46 ± 0.124 | 0.78 ± 0.063 | 0.74 ± 0.058 |
| 4 | Age Distribution Tail Emphasis | While still prioritizing all, slightly focus on both the youngest and oldest by age. | 1.00 | -0.02 ± 0.069 | 0.52 ± 0.111 | 1.00 ± 0.018 | 0.92 ± 0.018 |
| 5 | Middle Income | While still prioritizing all, slightly prefer the income bracket bounds for the middle 40% of the population. | 1.00 | -0.08 ± 0.067 | 0.65 ± 0.097 | 0.77 ± 0.057 | 0.88 ± 0.044 |
| 6 | Non-Phone Owner | While still prioritizing all, slightly favor those women who do not own their own phone. | 1.00 | -0.07 ± 0.067 | 0.50 ± 0.114 | 0.87 ± 0.028 | 0.97 ± 0.019 |
| 7 | Low Income + Younger Bias | While still prioritizing all, slightly prioritize impoverished younger mothers by combining the distributions of 'age' and 'education'. | 1.00 | 0.03 ± 0.075 | 0.40 ± 0.134 | 0.95 ± 0.014 | 0.98 ± 0.006 |
| 8 | Marathi Speakers + Middle Aged | While still prioritizing all, slightly focus on those Marathi-speakers with middle-aged mothers. | 1.00 | -0.00 ± 0.074 | 0.51 ± 0.115 | 0.82 ± 0.036 | 0.89 ± 0.023 |
| 9 | Early and Late Workers | While still prioritizing all, slightly emphasize beneficiaries who likely work early in the morning and late at night. | 1.00 | -0.11 ± 0.065 | 0.69 ± 0.093 | 0.81 ± 0.038 | 0.87 ± 0.038 |
| 10 | Critical Low Income | While still prioritizing all, slightly weight the lowest income_bracket groups, the absolute lowest earners in the population. | 1.00 | -0.08 ± 0.068 | 0.60 ± 0.104 | 0.93 ± 0.022 | 0.93 ± 0.015 |
| 11 | Early Morning Call + NGO Registered | While still prioritizing all, slightly advantage those who prefer being called before 10:30am 'slot' and are registered at an NGO. | 1.00 | -0.12 ± 0.065 | 0.70 ± 0.085 | 0.87 ± 0.037 | 0.96 ± 0.026 |
| 12 | Morning Call + NGO Registered | While still prioritizing all, slightly advantage those who prefer being called between 10:30am-12:30pm and are registered at an NGO. | 1.00 | -0.07 ± 0.066 | 0.52 ± 0.109 | 0.94 ± 0.020 | 0.87 ± 0.025 |
| 13 | Afternoon Call + NGO Registered | While still prioritizing all, slightly advantage those who prefer being called between 12:30pm-3:30pm and are registered at an NGO. | 1.00 | -0.11 ± 0.062 | 0.72 ± 0.085 | 0.92 ± 0.023 | 0.91 ± 0.024 |
| 14 | Evening Call + NGO Registered | While still prioritizing all, slightly advantage those who prefer being called after 7PM 'slot' registered at an NGO. | 1.00 | -0.06 ± 0.070 | 0.55 ± 0.111 | 0.67 ± 0.057 | 0.90 ± 0.031 |
| 15 | Technically Challenged | While still prioritizing all, infer technical challenges in reaching the phone that could indicate 'at-risk' beneficiaries and give slight preference. | 1.00 | -0.14 ± 0.060 | 0.64 ± 0.097 | 0.79 ± 0.049 | 0.92 ± 0.019 |
| **Avg.** | **Average MNR** | Average MNR for given 16 tasks | **1.00** | **-0.06 ± 0.017** | **0.57 ± 0.027** | **0.85 ± 0.008** | **0.92 ± 0.006** |

## K.2    Statistical Tests

Table 5: One-tailed t-tests comparing MNR scores against the null hypothesis that the mean MNR scores are less than or equal to the comparison group.

| Label | Default | DLM (No Reflection) | DLM (Reflection) | DLM (No Reflection) >Default? | DLM (Reflection) >Default? | DLM (Reflection) >DLM (No Reflection)? |
|---|---|---|---|---|---|---|
| Older Bias | 0.41 ± 0.130 | 0.91 ± 0.029 | 0.99 ± 0.005 | **p <0.001** | **p <0.001** | **p = 0.004** |
| Low Income | 0.51 ± 0.122 | 0.83 ± 0.046 | 0.97 ± 0.014 | **p = 0.007** | **p <0.001** | **p = 0.002** |
| Hindi Speakers | 0.71 ± 0.093 | 0.79 ± 0.058 | 0.98 ± 0.042 | p = 0.233 | **p = 0.004** | **p = 0.004** |
| Low Education Bias | 0.46 ± 0.124 | 0.78 ± 0.063 | 0.74 ± 0.058 | **p = 0.011** | **p = 0.021** | p = 0.680 |
| Age Distribution Tail Emphasis | 0.52 ± 0.111 | 1.00 ± 0.018 | 0.92 ± 0.018 | **p <0.001** | **p <0.001** | p = 0.999 |
| Middle Income | 0.65 ± 0.097 | 0.77 ± 0.057 | 0.88 ± 0.044 | p = 0.144 | **p = 0.016** | p = 0.064 |
| Non-Phone Owner | 0.50 ± 0.114 | 0.87 ± 0.028 | 0.97 ± 0.019 | **p <0.001** | **p <0.001** | **p = 0.002** |
| Low Income + Younger Bias | 0.40 ± 0.134 | 0.95 ± 0.014 | 0.98 ± 0.006 | **p <0.001** | **p <0.001** | **p = 0.025** |
| Marathi Speakers + Middle Aged | 0.51 ± 0.115 | 0.82 ± 0.036 | 0.89 ± 0.023 | **p = 0.005** | **p <0.001** | p = 0.051 |
| Early and Late Workers | 0.69 ± 0.093 | 0.81 ± 0.038 | 0.87 ± 0.038 | p = 0.117 | **p = 0.037** | p = 0.133 |
| Critical Low Income | 0.60 ± 0.104 | 0.93 ± 0.022 | 0.93 ± 0.015 | **p = 0.001** | **p <0.001** | p = 0.500 |
| Early Morning Call + NGO Registered | 0.70 ± 0.085 | 0.87 ± 0.037 | 0.96 ± 0.026 | **p = 0.034** | **p = 0.002** | **p = 0.024** |
| Morning Call + NGO Registered | 0.52 ± 0.109 | 0.94 ± 0.020 | 0.87 ± 0.025 | **p <0.001** | **p = 0.001** | p = 0.985 |
| Afternoon Call + NGO Registered | 0.72 ± 0.085 | 0.92 ± 0.023 | 0.91 ± 0.024 | **p = 0.012** | **p = 0.016** | p = 0.618 |
| Evening Call + NGO Registered | 0.55 ± 0.111 | 0.67 ± 0.057 | 0.90 ± 0.031 | p = 0.169 | **p = 0.001** | **p <0.001** |
| Technically Challenged | 0.64 ± 0.097 | 0.79 ± 0.049 | 0.92 ± 0.019 | p = 0.085 | **p = 0.003** | **p = 0.007** |

Here we include the full prompt list for each task, base reward functions, as well as the numerical results given in Figure 2. We show the full set of statistical tests in Table 5.

# L    Chain-of-Thought Experiments

We note that methods that improve LLM reasoning, such as chain-of-thought (CoT), may be incorporated to improve the base LLM reasoning in the DLM pipeline. Below, we include results using chain-of-thought reasoning for our model outputs; while we did not find that CoT significantly improved reward, we nevertheless emphasize the potential for additional LLM techniques to improve DLM outputs.

# M    Sample Reward Reflection Output

Here we provide the full output of a sample reward reflection procedure, in addition to the corresponding selected LLM candidate reward.

## M.1    Task: Older Bias

We add a full reflection output string for the Idx. 0: Older Bias task below. We make several observations on the proposed rewards and selected best candidate index. First, we note that the original base function uses only `agent_feats[11]`, which is the binary feature for the oldest age group Age 51-60. We see that the first proposed reward (Index 0) proposes using the top three oldest age groups, `agent_feats[9]`,`agent_feats[10]`,`agent_feats[11]`, while the second proposed reward (Index 1) uses only the top two oldest age groups `agent_feats[9]`,`agent_feats[10]`. We additionally observe, in the state-feature distributions, that the Index 1 reward observes a higher percentage of positive state accumulated for the oldest age group Ages 51-60, with 31.65% accumulated state vs 28.17% for Index 0. Finally, the LLM selects reward at Index 1 as the top candidate reward, demonstrating that there is some reasonable similarity between the selection of the LLM and the intuitively preferred reward function. Additionally, in this case, the LLM reflection was able to help guide the selection of reward functions that prioritize the very oldest age groups, demonstrating some positive effect on feature extraction through reflection.

Table 6: Full prompts and numerical results **with chain-of-thought (CoT)** reasoning for DLM.

| idx | Label | Full Prompt | Base (topline) | No Action | Default | DLM with CoT (No Reflection) | DLM with CoT (Reflection) |
|---|---|---|---|---|---|---|---|
| 0 | Older Bias | While still prioritizing all, slightly focus on the oldest by age distribution. | 1.00 | -0.02 ± 0.072 | 0.41 ± 0.130 | 0.89 ± 0.034 | 0.95 ± 0.023 |
| 1 | Low Income | While still prioritizing all, slightly focus on the bottom 12.5% of the income_bracket distribution. | 1.00 | -0.02 ± 0.073 | 0.51 ± 0.122 | 0.71 ± 0.062 | 0.62 ± 0.084 |
| 2 | Hindi Speakers | While still prioritizing all, slightly focus on those who speak Hindi. | 1.00 | -0.12 ± 0.067 | 0.71 ± 0.093 | 0.79 ± 0.055 | 0.94 ± 0.032 |
| 3 | Low Education Bias | While still prioritizing all, slightly weight those who have had low education. | 1.00 | -0.00 ± 0.073 | 0.46 ± 0.124 | 0.79 ± 0.059 | 0.72 ± 0.065 |
| 4 | Age Distribution Tail Emphasis | While still prioritizing all, slightly focus on both the youngest and oldest by age. | 1.00 | -0.02 ± 0.069 | 0.52 ± 0.111 | 0.64 ± 0.062 | 0.78 ± 0.044 |
| 5 | Middle Income | While still prioritizing all, slightly prefer the income bracket bounds for the middle 40% of the population. | 1.00 | -0.08 ± 0.067 | 0.65 ± 0.097 | 0.69 ± 0.072 | 0.66 ± 0.083 |
| 6 | Non-Phone Owner | While still prioritizing all, slightly favor those women who do not own their own phone. | 1.00 | -0.07 ± 0.067 | 0.50 ± 0.114 | 0.92 ± 0.029 | 0.85 ± 0.038 |
| 7 | Low Income + Younger Bias | While still prioritizing all, slightly prioritize impoverished younger mothers by combining the distributions of 'age' and 'education'. | 1.00 | 0.03 ± 0.075 | 0.40 ± 0.134 | 0.91 ± 0.021 | 0.96 ± 0.019 |
| 8 | Marathi Speakers + Middle Aged | While still prioritizing all, slightly focus on those Marathi-speakers with middle-aged mothers. | 1.00 | -0.00 ± 0.074 | 0.51 ± 0.115 | 0.88 ± 0.033 | 0.91 ± 0.033 |
| 9 | Early and Late Workers | While still prioritizing all, slightly emphasize beneficiaries who likely work early in the morning and late at night. | 1.00 | -0.11 ± 0.065 | 0.69 ± 0.093 | 0.93 ± 0.035 | 0.94 ± 0.024 |
| 10 | Critical Low Income | While still prioritizing all, slightly weight the lowest income_bracket groups, the absolute lowest earners in the population. | 1.00 | -0.08 ± 0.068 | 0.60 ± 0.104 | 0.85 ± 0.041 | 0.83 ± 0.041 |
| 11 | Early Morning Call + NGO Registered | While still prioritizing all, slightly advantage those who prefer being called before 10:30am 'slot' and are registered at an NGO. | 1.00 | -0.12 ± 0.065 | 0.70 ± 0.085 | 0.94 ± 0.027 | 0.94 ± 0.027 |
| 12 | Morning Call + NGO Registered | While still prioritizing all, slightly advantage those who prefer being called between 10:30am-12:30pm and are registered at an NGO. | 1.00 | -0.07 ± 0.066 | 0.52 ± 0.109 | 0.93 ± 0.026 | 0.92 ± 0.031 |
| 13 | Afternoon Call + NGO Registered | While still prioritizing all, slightly advantage those who prefer being called between 12:30pm-3:30pm and are registered at an NGO. | 1.00 | -0.11 ± 0.062 | 0.72 ± 0.085 | 0.76 ± 0.052 | 0.82 ± 0.041 |
| 14 | Evening Call + NGO Registered | While still prioritizing all, slightly advantage those who prefer being called after 7PM 'slot' registered at an NGO. | 1.00 | -0.06 ± 0.070 | 0.55 ± 0.111 | 0.83 ± 0.046 | 0.78 ± 0.045 |
| 15 | Technically Challenged | While still prioritizing all, infer technical challenges in reaching the phone that could indicate 'at-risk' beneficiaries and give slight preference. | 1.00 | -0.14 ± 0.060 | 0.64 ± 0.097 | 0.80 ± 0.045 | 0.90 ± 0.037 |

```
My goal was to create a Python reward function for RL in resource allocation,
with the objective of: While still prioritizing all, slightly focus on the
oldest by age distribution. I tried several reward functions for this task.
Below, I have the given reward function, and the corresponding distribution of
reward achieved across 44 agent features. A description of the features is as
follows:
Index Name DataType
[sensitive features hidden]
7. Ages 10-20 - Binary
8. Ages 21-30 - Binary
9. Ages 31-40 - Binary
10. Ages 41-50 - Binary
11. Ages 51-60 - Binary
12. Speaks Hindi - Binary
13. Speaks Marathi - Binary
14. Speaks Gujurati - Binary
15. Speaks Kannada - Binary
16. Education level 1/7 -- illiterate - Binary
17. Education level 2/7 -- 1-5th Grade Completed - Binary
18. Education level 3/7 -- 6-9th Grade Completed - Binary
19. Education level 4/7 -- 10th Grade Passed - Binary
20. Education level 5/7 -- 12th Grade Passed - Binary
21. Education level 6/7 -- Graduate - Binary
22. Education level 7/7 -- Post graduate - Binary
23. Phone owner 0 (e.g., woman) - Binary
24. Phone owner 1 (e.g., husband) - Binary
25. Phone owner 2 (e.g., family) - Binary
26. To be called from 8:30am-10:30am - Binary
27. To be called from 10:30am-12:30pm - Binary
```

```
28. To be called from 12:30pm-3:30pm - Binary
29. To be called from 3:30pm-5:30pm - Binary
30. To be called from 5:30pm-7:30pm - Binary
31. To be called from 7:30pm-9:30pm - Binary
32. NGO - Binary
33. ARMMAN - Binary
34. PHC - Binary
35. Income bracket -1 (no income) - Binary
36. Income bracket 1 (e.g., 0-5000) - Binary
37. Income bracket 2 (e.g., 5001-10000) - Binary
38. Income bracket 3 (e.g., 10001-15000) - Binary
39. Income bracket 4 (e.g., 15001-20000) - Binary
40. Income bracket 5 (e.g., 20001-25000) - Binary
41. Income bracket 6 (e.g., 25001-30000) - Binary
42. Income bracket 7 (e.g., 30000-999999) - Binary

Below are the reward functions I used and their corresponding reward
distributions:

Index 0:
Reward Function: 3 * (state) + 4 * ( (state) * (agent_feats[9] or
agent_feats[10] or agent_feats[11]) )
Reflection:
'

[sensitive features hidden]

Category: Ages
Ages 10-20: 18.18%
Ages 21-30: 27.75%
Ages 31-40: 22.95%
Ages 41-50: 2.95%
Ages 51-60: 28.17%

Category: Income
Income bracket -1 (no income): 0.00%
Income bracket 1 (e.g., 0-5000): 9.13%
Income bracket 2 (e.g., 5001-10000): 55.81%
Income bracket 3 (e.g., 10001-15000): 23.35%
Income bracket 4 (e.g., 15001-20000): 5.85%
Income bracket 5 (e.g., 20001-25000): 5.87%
Income bracket 6 (e.g., 25001-30000): 0.00%
Income bracket 7 (e.g., 30000-999999): 0.00%

Category: Calling Times
8:30am-10:30am: 7.45%
10:30am-12:30pm: 4.68%
12:30pm-3:30pm: 53.17%
3:30pm-5:30pm: 13.74%
5:30pm-7:30pm: 9.48%
7:30pm-9:30pm: 11.48%

Category: Education Levels
Illiterate: 4.75%
1-5th Grade Completed: 6.22%
6-9th Grade Completed: 31.70%
10th Grade Passed: 20.22%
12th Grade Passed: 37.10%
Graduate: 0.00%
Post graduate: 0.00%

Category: Languages Spoken
Speaks Hindi: 35.23%
Speaks Marathi: 64.77%
Speaks Gujurati: 0.00%
Speaks Kannada: 0.00%

Category: Phone Owners
Phone owner - Woman: 91.23%
Phone owner - Husband: 5.72%
Phone owner - Family: 3.05%

Category: Organizations
NGO: 75.58%
ARMMAN: 24.42%
PHC: 0.00%'

Index 1:
Reward Function: state * 0.1 + 2 * state * (agent_feats[10] or agent_feats[11])
Reflection:
```

```
,
[sensitive features hidden]

Category: Ages
Ages 10-20: 17.42%
Ages 21-30: 26.24%
Ages 31-40: 21.78%
Ages 41-50: 2.90%
Ages 51-60: 31.65%

Category: Income
Income bracket -1 (no income): 0.00%
Income bracket 1 (e.g., 0-5000): 8.77%
Income bracket 2 (e.g., 5001-10000): 58.05%
Income bracket 3 (e.g., 10001-15000): 21.70%
Income bracket 4 (e.g., 15001-20000): 5.73%
Income bracket 5 (e.g., 20001-25000): 5.75%
Income bracket 6 (e.g., 25001-30000): 0.00%
Income bracket 7 (e.g., 30000-999999): 0.00%

Category: Calling Times
8:30am-10:30am: 7.32%
10:30am-12:30pm: 4.39%
12:30pm-3:30pm: 55.10%
3:30pm-5:30pm: 12.90%
5:30pm-7:30pm: 8.73%
7:30pm-9:30pm: 11.55%

Category: Education Levels
Illiterate: 4.56%
1-5th Grade Completed: 5.86%
6-9th Grade Completed: 30.28%
10th Grade Passed: 19.07%
12th Grade Passed: 40.24%
Graduate: 0.00%
Post graduate: 0.00%

Category: Languages Spoken
Speaks Hindi: 33.67%
Speaks Marathi: 66.33%
Speaks Gujurati: 0.00%
Speaks Kannada: 0.00%

Category: Phone Owners
Phone owner - Woman: 91.34%
Phone owner - Husband: 5.76%
Phone owner - Family: 2.89%

Category: Organizations
NGO: 76.50%
ARMMAN: 23.50%
PHC: 0.00%'

Based on the above reward distributions and the given goal: While still
prioritizing all, slightly focus on the oldest by age distribution., please
identify the index of the most effective reward function. Provide your answer
EXACTLY IN the following format: 'The best reward function is at index:
[INDEX]'.
```

```
The best reward function is at index:  1
```

## M.2   Marathi Speakers + Middle Aged

We next analyze a sample reward reflection for the Idx. 8: Marathi Speakers + Middle aged task below. Observe that the base reward function in Table 3 uses the features `agent_feats[13]`, the binary Marathi-speaking feature, and `agent_feats[9]`,`agent_feats[10]`, the middle ages in the age range Ages 31-40 and Ages 41-50. Below, we observe that the first proposed reward includes these exact features, while the second proposed reward only includes one of the middle-income features `agent_feats[9]`. However, there is a key difference in the feature *weightings* applied in both reward functions, where the first proposed reward function weights the prioritized features much more with scalar multipliers than the first proposed function. In this case, the LLM selects the first proposed reward function, which also happens to be closer to the scaling used in the original Base reward function. While this is not known to the LLM, the first proposed reward function also results

in a greater accumulate state percentage in the relevant middle-age feature groups, a possible reason for the LLM selecting the first reward, thus demonstrating an example of reflection aiding in tuning feature weighting in reward functions.

```
My goal was to create a Python reward function for RL in resource allocation,
with the objective of: While still prioritizing all, slightly focus on those
Marathi-speakers with middle-aged mothers. I tried several reward functions for
this task. Below, I have the given reward function, and the corresponding
distribution of reward achieved across 44 agent features. A description of the
features is as follows:
Index Name DataType
[sensitive features hidden]
7. Ages 10-20 - Binary
8. Ages 21-30 - Binary
9. Ages 31-40 - Binary
10. Ages 41-50 - Binary
11. Ages 51-60 - Binary
12. Speaks Hindi - Binary
13. Speaks Marathi - Binary
14. Speaks Gujurati - Binary
15. Speaks Kannada - Binary
16. Education level 1/7 -- illiterate - Binary
17. Education level 2/7 -- 1-5th Grade Completed - Binary
18. Education level 3/7 -- 6-9th Grade Completed - Binary
19. Education level 4/7 -- 10th Grade Passed - Binary
20. Education level 5/7 -- 12th Grade Passed - Binary
21. Education level 6/7 -- Graduate - Binary
22. Education level 7/7 -- Post graduate - Binary
23. Phone owner 0 (e.g., woman) - Binary
24. Phone owner 1 (e.g., husband) - Binary
25. Phone owner 2 (e.g., family) - Binary
26. To be called from 8:30am-10:30am - Binary
27. To be called from 10:30am-12:30pm - Binary
28. To be called from 12:30pm-3:30pm - Binary
29. To be called from 3:30pm-5:30pm - Binary
30. To be called from 5:30pm-7:30pm - Binary
31. To be called from 7:30pm-9:30pm - Binary
32. NGO - Binary
33. ARMMAN - Binary
34. PHC - Binary
35. Income bracket -1 (no income) - Binary
36. Income bracket 1 (e.g., 0-5000) - Binary
37. Income bracket 2 (e.g., 5001-10000) - Binary
38. Income bracket 3 (e.g., 10001-15000) - Binary
39. Income bracket 4 (e.g., 15001-20000) - Binary
40. Income bracket 5 (e.g., 20001-25000) - Binary
41. Income bracket 6 (e.g., 25001-30000) - Binary
42. Income bracket 7 (e.g., 30000-999999) - Binary

Below are the reward functions I used and their corresponding reward
distributions:

Index 0:
Reward Function: state * 0.1 + 3.5 * state * ((agent_feats[9] or
agent_feats[10]) and agent_feats[13])
Reflection:
'
[sensitive features hidden]

Category: Ages
Ages 10-20: 4.02%
Ages 21-30: 12.04%
Ages 31-40: 82.79%
Ages 41-50: 1.16%
Ages 51-60: 0.00%

Category: Income
Income bracket -1 (no income): 0.00%
Income bracket 1 (e.g., 0-5000): 2.49%
Income bracket 2 (e.g., 5001-10000): 62.28%
Income bracket 3 (e.g., 10001-15000): 32.25%
Income bracket 4 (e.g., 15001-20000): 2.43%
Income bracket 5 (e.g., 20001-25000): 0.56%
Income bracket 6 (e.g., 25001-30000): 0.00%
Income bracket 7 (e.g., 30000-999999): 0.00%

Category: Calling Times
8:30am-10:30am: 5.45%
10:30am-12:30pm: 13.50%
```

```
12:30pm-3:30pm: 32.12%
3:30pm-5:30pm: 17.42%
5:30pm-7:30pm: 1.74%
7:30pm-9:30pm: 29.77%

Category: Education Levels
Illiterate: 1.13%
1-5th Grade Completed: 18.09%
6-9th Grade Completed: 32.36%
10th Grade Passed: 30.49%
12th Grade Passed: 14.38%
Graduate: 0.59%
Post graduate: 2.96%

Category: Languages Spoken
Speaks Hindi: 16.86%
Speaks Marathi: 83.14%
Speaks Gujurati: 0.00%
Speaks Kannada: 0.00%

Category: Phone Owners
Phone owner - Woman: 95.22%
Phone owner - Husband: 3.02%
Phone owner - Family: 1.76%

Category: Organizations
NGO: 92.87%
ARMMAN: 7.13%
PHC: 0.00%'

Index 1:
Reward Function: 2 * state + 2 * (state and (agent_feats[9] and
agent_feats[13]))
Reflection:
'
[sensitive features hidden]

Category: Ages
Ages 10-20: 4.33%
Ages 21-30: 12.53%
Ages 31-40: 81.90%
Ages 41-50: 1.23%
Ages 51-60: 0.00%

Category: Income
Income bracket -1 (no income): 0.00%
Income bracket 1 (e.g., 0-5000): 2.48%
Income bracket 2 (e.g., 5001-10000): 63.04%
Income bracket 3 (e.g., 10001-15000): 31.35%
Income bracket 4 (e.g., 15001-20000): 2.47%
Income bracket 5 (e.g., 20001-25000): 0.66%
Income bracket 6 (e.g., 25001-30000): 0.00%
Income bracket 7 (e.g., 30000-999999): 0.00%

Category: Calling Times
8:30am-10:30am: 5.58%
10:30am-12:30pm: 12.93%
12:30pm-3:30pm: 34.84%
3:30pm-5:30pm: 17.14%
5:30pm-7:30pm: 1.81%
7:30pm-9:30pm: 27.70%

Category: Education Levels
Illiterate: 1.27%
1-5th Grade Completed: 17.84%
6-9th Grade Completed: 34.92%
10th Grade Passed: 29.50%
12th Grade Passed: 12.58%
Graduate: 0.55%
Post graduate: 3.35%

Category: Languages Spoken
Speaks Hindi: 18.26%
Speaks Marathi: 81.74%
Speaks Gujurati: 0.00%
Speaks Kannada: 0.00%

Category: Phone Owners
Phone owner - Woman: 94.85%
Phone owner - Husband: 3.23%
Phone owner - Family: 1.91%
```

```
Category: Organizations
NGO: 92.32%
ARMMAN: 7.68%
PHC: 0.00%'

Based on the above reward distributions and the given goal: While still
prioritizing all, slightly focus on those Marathi-speakers with middle-aged
mothers., please identify the index of the most effective reward function.
Provide your answer EXACTLY IN the following format: 'The best reward function
is at index: [INDEX]'.
```

```
The best reward function is at index:  0
```

## M.3   Technically Challenged

In the final example, we compare a sample of reward reflection for the challenging, ambiguous
"Technically Challenged" prompt. The ground truth Base reward for this task considers only the
features of phone ownership, interpreting "Technically Challenged" as those women who do not
own their own phone (using features `agent_feats[24]`, `agent_feats[25]`. In this case, both
proposed reward functions have low precision, including auxillary features that are not directly
relevant to the Base reward. We observe, in this case, that although the second proposed reward, at
Index 1, has utilized only one of the relevant features `agent_feats[25]`, it is ultimately selected in
the reflection stage. For this highly ambiguous tasks, additional external input may be required, as
we observe in this case that reflection does not align with what we may desire given the known Base
reward function.

```
My goal was to create a Python reward function for RL in resource allocation,
with the objective of: While still prioritizing all, infer technical challenges
in reaching the phone that could indicate 'at-risk' beneficiaries and give
slight preference. I tried several reward functions for this task. Below, I have
the given reward function, and the corresponding distribution of reward achieved
across 44 agent features. A description of the features is as follows:
Index Name DataType
[sensitive features hidden]
7. Ages 10-20 - Binary
8. Ages 21-30 - Binary
9. Ages 31-40 - Binary
10. Ages 41-50 - Binary
11. Ages 51-60 - Binary
12. Speaks Hindi - Binary
13. Speaks Marathi - Binary
14. Speaks Gujurati - Binary
15. Speaks Kannada - Binary
16. Education level 1/7 -- illiterate - Binary
17. Education level 2/7 -- 1-5th Grade Completed - Binary
18. Education level 3/7 -- 6-9th Grade Completed - Binary
19. Education level 4/7 -- 10th Grade Passed - Binary
20. Education level 5/7 -- 12th Grade Passed - Binary
21. Education level 6/7 -- Graduate - Binary
22. Education level 7/7 -- Post graduate - Binary
23. Phone owner 0 (e.g., woman) - Binary
24. Phone owner 1 (e.g., husband) - Binary
25. Phone owner 2 (e.g., family) - Binary
26. To be called from 8:30am-10:30am - Binary
27. To be called from 10:30am-12:30pm - Binary
28. To be called from 12:30pm-3:30pm - Binary
29. To be called from 3:30pm-5:30pm - Binary
30. To be called from 5:30pm-7:30pm - Binary
31. To be called from 7:30pm-9:30pm - Binary
32. NGO - Binary
33. ARMMAN - Binary
34. PHC - Binary
35. Income bracket -1 (no income) - Binary
36. Income bracket 1 (e.g., 0-5000) - Binary
37. Income bracket 2 (e.g., 5001-10000) - Binary
38. Income bracket 3 (e.g., 10001-15000) - Binary
39. Income bracket 4 (e.g., 15001-20000) - Binary
40. Income bracket 5 (e.g., 20001-25000) - Binary
41. Income bracket 6 (e.g., 25001-30000) - Binary
42. Income bracket 7 (e.g., 30000-999999) - Binary

Below are the reward functions I used and their corresponding reward
```

```
distributions:

Index 0:
Reward Function: state * 0.1 + 2 * state * ((agent_feats[9] or agent_feats[10])
and (agent_feats[11] or agent_feats[24] or agent_feats[25]))
Reflection:
'

[sensitive features hidden]

Category: Ages
Ages 10-20: 5.38%
Ages 21-30: 84.35%
Ages 31-40: 6.31%
Ages 41-50: 3.96%
Ages 51-60: 0.00%

Category: Income
Income bracket -1 (no income): 0.00%
Income bracket 1 (e.g., 0-5000): 6.98%
Income bracket 2 (e.g., 5001-10000): 44.30%
Income bracket 3 (e.g., 10001-15000): 43.07%
Income bracket 4 (e.g., 15001-20000): 4.83%
Income bracket 5 (e.g., 20001-25000): 0.82%
Income bracket 6 (e.g., 25001-30000): 0.00%
Income bracket 7 (e.g., 30000-999999): 0.00%

Category: Calling Times
8:30am-10:30am: 10.20%
10:30am-12:30pm: 3.16%
12:30pm-3:30pm: 9.29%
3:30pm-5:30pm: 19.40%
5:30pm-7:30pm: 21.35%
7:30pm-9:30pm: 36.59%

Category: Education Levels
Illiterate: 2.42%
1-5th Grade Completed: 20.64%
6-9th Grade Completed: 27.74%
10th Grade Passed: 24.80%
12th Grade Passed: 4.62%
Graduate: 1.66%
Post graduate: 18.13%

Category: Languages Spoken
Speaks Hindi: 37.45%
Speaks Marathi: 62.55%
Speaks Gujurati: 0.00%
Speaks Kannada: 0.00%

Category: Phone Owners
Phone owner - Woman: 34.69%
Phone owner - Husband: 31.75%
Phone owner - Family: 33.56%

Category: Organizations
NGO: 43.54%
ARMMAN: 56.46%
PHC: 0.00%'

Index 1:
Reward Function: (5 * agent_feats[7] + 4 * agent_feats[8] + 3 * agent_feats[14]
+ 2 * agent_feats[25] + 1) * state
Reflection:
'

[sensitive features hidden]

Category: Ages
Ages 10-20: 5.21%
Ages 21-30: 84.78%
Ages 31-40: 6.00%
Ages 41-50: 4.01%
Ages 51-60: 0.00%

Category: Income
Income bracket -1 (no income): 0.00%
Income bracket 1 (e.g., 0-5000): 6.88%
Income bracket 2 (e.g., 5001-10000): 44.74%
Income bracket 3 (e.g., 10001-15000): 42.90%
Income bracket 4 (e.g., 15001-20000): 4.65%
```

```
Income bracket 5 (e.g., 20001-25000): 0.83%
Income bracket 6 (e.g., 25001-30000): 0.00%
Income bracket 7 (e.g., 30000-999999): 0.00%

Category: Calling Times
8:30am-10:30am: 10.03%
10:30am-12:30pm: 2.98%
12:30pm-3:30pm: 9.01%
3:30pm-5:30pm: 19.40%
5:30pm-7:30pm: 21.43%
7:30pm-9:30pm: 37.16%

Category: Education Levels
Illiterate: 2.30%
1-5th Grade Completed: 21.66%
6-9th Grade Completed: 27.61%
10th Grade Passed: 24.72%
12th Grade Passed: 4.53%
Graduate: 1.53%
Post graduate: 17.67%

Category: Languages Spoken
Speaks Hindi: 36.94%
Speaks Marathi: 63.06%
Speaks Gujurati: 0.00%
Speaks Kannada: 0.00%

Category: Phone Owners
Phone owner - Woman: 33.76%
Phone owner - Husband: 32.50%
Phone owner - Family: 33.74%

Category: Organizations
NGO: 43.09%
ARMMAN: 56.91%
PHC: 0.00%'

Based on the above reward distributions and the given goal: While still
prioritizing all, infer technical challenges in reaching the phone that could
indicate 'at-risk' beneficiaries and give slight preference., please identify
the index of the most effective reward function. Provide your answer EXACTLY IN
the following format: 'The best reward function is at index: [INDEX]'.
```

```
The best reward function is at index:  1
```

# N    Generated Reward Functions

