# OpenReview forum: "A Decision-Language Model (DLM) for Dynamic Restless Multi-Armed Bandit Tasks in Public Health"
_NeurIPS.cc/2024/Conference — NeurIPS 2024 poster_

### Official Review · Reviewer_zXh7 · 2024-07-01

**Soundness:** 3
**Presentation:** 3
**Contribution:** 3
**Rating:** 7
**Confidence:** 3

**Summary:**

The paper titled "A Decision-Language Model (DLM) for Dynamic Restless Multi-Armed Bandit Tasks in Public Health" introduces a novel method for improving public health resource allocation. By combining Restless Multi-Armed Bandit (RMAB) models with the interpretive power of Large Language Models (LLMs), the authors have created a system that can adaptively fine-tune health policies based on human-language commands. This innovation allows for more flexible and responsive policy adjustments, crucial for addressing the changing needs of public health programs.
A significant contribution of this work is showing how LLMs can generate and refine reward functions for RMABs, enhancing decision-making in resource-limited settings. The authors demonstrate this through a collaboration with ARMMAN, an Indian non-profit focused on maternal health. Their simulations reveal that the DLM can significantly improve the allocation of health workers, leading to better engagement and outcomes. This approach promises to make public health interventions more effective and adaptable, closely aligning with the evolving needs of the community.

**Strengths:**

Originality:
I think the originality of this paper quite impressive. The way it combines Large Language Models (LLMs) with Restless Multi-Armed Bandits (RMABs) to dynamically fine-tune public health policies is really innovative. Using human-language commands to adjust these policies is a clever idea that bridges advanced AI techniques and practical decision-making in a unique way. This creative approach brings a fresh perspective to public health, making it much more adaptable and responsive to changing needs.
Quality:
The quality of the research really stands out. The authors did a good job detailing their methodology, from the reward proposal loop to the simulation stages and the reflection mechanism for refining reward functions. For example, they explain how LLMs interpret policy preferences and generate reward functions, which are then fine-tuned through simulations. The experiments are well-thought-out and use real-world data from ARMMAN, which adds a lot of credibility. The results are impressive, demonstrating DLM can achieve near human-level performance. The authors also provide a comprehensive analysis, comparing their model to baseline methods using clear performance metrics. This thorough validation highlights the potential for further real-world applications, showcasing how the approach can dynamically adjust policies to meet evolving public health needs.
Clarity:
The paper is generally well-organized and easy to follow. The tables and figures do a great job of illustrating the key concepts and results. However, some parts could be more engaging and less technical, making it easier for a wider audience to understand. Simplifying some of the technical language and adding short summaries at the end of sections would help readers quickly get the main points without getting lost in the details. For example, a brief recap of the key findings at the end of the results section would be really helpful.
Significance:
Despite being based on simulations, this work shows great potential for real-world impact. The DLM can dynamically adjust RMAB policies to meet changing public health needs, which is incredibly important. The evaluation on real-world data show that this approach isn't just theoretical but has practical relevance. The findings suggest that the DLM could significantly improve how resources are allocated and how effective policies are, which is very promising. Being able to dynamically prioritize different demographic groups or regions based on evolving needs can lead to more targeted and efficient use of resources. This approach could have wide applications in various public health interventions.

**Weaknesses:**

One major limitation of the paper is its reliance on a simulated environment for validation, which is understandable given the complexity of real-world testing. While simulations using real-world data, the findings would be significantly strengthened by real-world trials. A practical next step would be to outline a detailed plan for field testing the Decision-Language Model (DLM) in actual public health settings. This would include partnerships with health organizations for pilot studies, addressing potential ethical and logistical challenges, and establishing metrics for real-world success.

**Questions:**

Is there any theoretical foundation you rely on for designing the prompts used in your model? Do you think referring to sociological theories or frameworks could help in designing more effective and contextually relevant prompts?
The paper mentions issues with ambiguous language prompts leading to misaligned policies. Do you have any specific methods or future plans to quantify and address these ambiguities to improve the model's reliability?

**Limitations:**

The simulations in this work are impressive, but to make a real-world impact, the authors need to address some key areas. These include transitioning to actual field trials, expanding language support, ensuring the model can handle larger datasets, and dealing with ambiguous prompts. It's also important to consider potential negative societal impacts like data privacy, bias, and the effects on vulnerable groups. By tackling these issues and outlining a clear path for real-world use, the authors can significantly enhance the practical and ethical value of their work.

---

> ### Author Rebuttal · Authors · 2024-08-06
>
> 1) Simulated Environment Reliance: “The paper relies on simulations for validation. Real-world trials would strengthen findings. A plan for field testing DLM in actual public health settings, including partnerships for pilot studies and addressing ethical and logistical challenges, is needed.”
>
> This work will be followed by a real world evaluation. Domain experts from the NGO played an integral role throughout the model iteration and design process. And we would also like to re-emphasize (see Section 5 and Appendix A)  that any subsequent real-world evaluations are only undertaken after securing ethics approval from the NGO's ethics board which is registered with the ICMR (Indian Council of Medical Research). Please see the global Note on Ethical Considerations above for more information. Obtaining approvals and designing and conducting real world evaluation require significant effort from the NGO and may take up to several months to complete due to complexities involved in real-world trials. Success in simulations are required for any subsequent approvals, and our simulations indicate substantial potential gains from the system (as demonstrated in this paper).
>
> 2) Field Trials and Testing: “Transitioning to real-world testing is essential. This includes expanding language support, handling larger datasets, and addressing ambiguous prompts.”
>
> We thank the reviewer for this comment and agree that these are critical steps for future envisioned deployment. We discuss potential strategies to handle ambiguous prompts below. As referenced in Sec. 5.6, we agree also that expanded testing across different languages is critical. This may intersect with ongoing research in multilingual LLM evaluation [10], especially in LLM reasoning from Indic languages [11] which are highly represented in the ARMMAN deployment regions. Please see above global comment for additional details in consideration of future envisioned real-world testing.
>
> 3) Ethical Considerations: “Address potential negative societal impacts such as data privacy, bias, and effects on vulnerable groups. Tackling these issues and outlining a clear path for real-world use will enhance the practical and ethical value of the work.”
>
> We thank the reviewer for the comment. We summarize the ethical considerations discussed in our paper in the global comment above, and highlight these key sections below.
>
> We include a discussion on societal impacts in Appendix A and a discussion on data usage in Appendix C. The real world dataset is completely anonymized by ARMMAN. Data exchange from ARMMAN to the researchers was regulated through clearly defined exchange protocols including anonymization, read-only researcher access, restricted use of data for research purposes only, and approval by the ARMMAN ethics review committee. Our method helps ARMMAN in training dynamic policies to support vulnerable groups and allows ARMMAN program managers to monitor state-feature distributions over underrepresented demographic groups (see Figure 1 description and Section 5). We will further emphasize above mentioned points on ethical considerations and societal impacts in the paper.
>
> 4) Prompt Design Foundation: “Is there a theoretical foundation for designing the prompts used in the model? Referring to sociological theories or frameworks could help in creating more effective and relevant prompts.”
>
> The prompts in our model are primarily designed to capture the practical, real-world scenarios that could be encountered through callership patterns amongst mothers in India and in collaboration with ARMMAN. While there isn't a direct theoretical foundation in prompt design, the prompts are inspired by common decision-making processes and challenges faced by ARMMAN. In addition, we design them in such a way that they become progressively more difficult to reason about, where we include a mixture of tasks that require just recovering one feature, multiple features, and inferring the correct due to ambiguity. Thus, a substantial amount of time was spent thinking about designing the prompt, although the current prompts could benefit from principled approaches.
>
> For example, we believe that integrating sociological theories or frameworks is a wonderful idea, and could indeed enhance the contextual relevance and effectiveness of prompts. For instance, a list of criteria such as the Social Determinants of Health [12] could be used to help guide the development of prompts that would  likely improve community needs and values. In addition, a large body of literature is now emerging along the direction of “prompt engineering” [13, 14], or finding methods for improving prompt efficacy.
>
> 5) Ambiguous Language Prompts: “The paper notes issues with ambiguous prompts leading to misaligned policies. Are there methods or plans to quantify and address these ambiguities to improve reliability?”
>
> One simple idea is to reject ambiguous prompts and ask for further clarifying information, which is tackled in other work that uses the disagreement between sampled  outputs of the model [15] to quantify ambiguity. This is a well-studied area [16, 17], and there should be several strategies available that effectively mitigate ambiguous questions.

---

> > ### Comment · Reviewer_zXh7 · 2024-08-07
> >
> > I have read the Rebuttal, thanks for answering my question.

---

### Official Review · Reviewer_R6Uk · 2024-07-08

**Soundness:** 3
**Presentation:** 3
**Contribution:** 3
**Rating:** 6
**Confidence:** 2

**Summary:**

This paper proposes using a decision-language model for restless multi-armed bandit tasks (RMAB) in the public health domain. The authors evaluated their method in a simulation environment developed from a real-world dataset. The authors conducted experiments with 16 different prompts and compared their approach with baselines, demonstrating the effectiveness of their method. Their study provided insights into the generated reward design for RMABs.

**Strengths:**

1. The authors evaluated their method in a simulation environment developed from a real-world dataset and provided comparisons in the DLM-generated model with the baseline model.
2. The author provides the parameters of the model training process, which helps other researchers reproduce the process.

**Weaknesses:**

1. Avoid citing sources in the Abstract
2. The approach provided in this paper lacks real-world validation.
3. This paper did not discuss the ethical implications of using AI for decision-making in health resource allocation.

**Questions:**

1. Why use the ARMMAN dataset exclusively in this study? What language is used in this dataset? Given that the test involves prompts in English, was there any preprocessing of the dataset required?
2. In some parts of the training process, it is crucial to explain why specific numbers are chosen. For instance, the downstream policy is trained for 5 epochs,  why 5?  why are 100 simulation steps selected?

**Limitations:**

The approach provided in this paper lacks real-world validation. This paper should also discuss the ethical implications of using AI for decision-making in health resource allocation.

---

> ### Author Rebuttal · Authors · 2024-08-06
>
> 1) Abstract Citations: “Avoid citing sources in the Abstract.”
>
> We thank the reviewer for the suggestion. We will remove this citation from the abstract. We include this citation at the end of the introduction.
>
> 2) Real-World Validation: “The approach provided in this paper lacks real-world validation.”
>
> This work will be followed by a real world evaluation. Domain experts from the NGO played an integral role throughout the model iteration and design process. And we would also like to re-emphasize (see Section 5 and Appendix A)  that any subsequent real-world evaluations are only undertaken after securing ethics approval from the NGO's ethics board which is registered with the ICMR (Indian Council of Medical Research). Please see the global Note on Ethical Considerations above for more information. Obtaining approvals and designing and conducting real world evaluation require significant effort from the NGO and may take up to several months to complete due to the complexities involved in real-world trials. Success in simulations are required for any subsequent approvals, and our simulations indicate substantial potential gains from the system (as demonstrated in this paper).
>
> 3) Training Process Explanation: “In some parts of the training process, it is crucial to explain why specific numbers are chosen. For instance, the downstream policy is trained for 5 epochs, why 5? Why are 100 simulation steps selected?”
>
> We thank the reviewer for this suggestion and will include additional details on the selection of training hyperparameters in the Appendix. Choices for these hyperparameters were made following quantitative assessments over a hyperparameter search which showed improved stability and low overfitting for the selected values for our given dataset. Our choices were additionally qualitatively guided by prior works studying network-based multi-armed bandit policy training, which similarly use 100 steps in buffer for simulated settings [9]. We will include this information on hyperparameter selection in the Appendix.
>
> 4) Ethical Implications: “This paper should also discuss the ethical implications of using AI for decision-making in health resource allocation.”
>
> We discuss this extensively in Appendix A, Social Impact Statement, as well as Appendix B, Dataset Description and Appendix C, Dataset Consent for Collection and Analysis. We summarize these points discussed in our paper in the global comment above, including an overview of the ethical considerations taken prior to and during this work with our partner NGO.
>
> 5) Dataset Usage: “Why use the ARMMAN dataset exclusively in this study? What language is used in this dataset? Given that the test involves prompts in English, was there any preprocessing of the dataset required?”
>
> We focus on the ARMMAN dataset because it is vetted, real-world data collected with consent ethically. Such a dataset, created from a study over 7,668 mothers, has the potential for great impact, helping us gain insight into key policy design techniques that can help improve the deployment of public health resources across millions of beneficiaries. Furthermore, the partner organization ARMMAN has an operational AI/RMAB deployment that works to allocate resources. The collected dataset is also rich and not sparse, allowing a diverse set of tasks to be tested within it. Thus, there are multiple important reasons to work with this dataset. The dataset was already collected in English. Thus, no preprocessing of the dataset was required. Due to these key strengths and benefits, we focus our analysis entirely on this highly valuable source of data.

---

> > ### Comment · Reviewer_R6Uk · 2024-08-08
> >
> > I thank the authors for answering my questions.

---

### Official Review · Reviewer_8Afe · 2024-07-12

**Soundness:** 3
**Presentation:** 3
**Contribution:** 3
**Rating:** 5
**Confidence:** 4

**Summary:**

Restless multi-armed bandits (RMAB) are effective for resource allocation in public health but lack adaptability to changing policies. Large Language Models (LLMs) have potential in healthcare for dynamic resource allocation through language prompts but are understudied in this area. This work introduces a Decision-Language Model (DLM) for RMABs to fine-tune resource allocation policies using language prompts. It proposes using LLMs to clarify policy preferences, propose reward functions, and iteratively refine these functions through simulations. Key contributions include pioneering the use of LLMs for adapting public health resource allocation and demonstrating near-human-level policy tuning in a maternal and child care task. They introduce a reward proposal loop that improves LLM-generated reward functions using feedback from restless multi-armed bandit (RMAB) simulations. This allows LLMs to iteratively refine reward designs to achieve specific, human-specified policy outcomes.

**Strengths:**

- It is very well written and fluent.
- They highlighted the comments and important parts which is really helpful to follow the context.
- The method is new in this application.

**Weaknesses:**

- The related works are not comprehensive. It should bring some works in healthcare from other approach and also the application of LLM in other healthcare examples.

- The policy and critic is not identified till in the algorithm 1 and that is confused the reader. Since from the beginning it seems, the policy is also LLM. It requires more clarification and adjustment.

- It is true that DLM (with reflection) works better but it is not a good fair comparison. Justifying why without reflection works good or bad is necessary.

- The baseline are not fair. It is true that might be other methods with LLM is not yet applied to this healthcare problem but to claim why they choose this algorithm, comparing with other algorithms that use LLM is necessary. They need to bring enough evidence for their selection.  In this case they need to bring baseline from the work with LLMs.

**Questions:**

- What are the feature z exactly? It is very helpful to bring some clear examples in section or the beginning of section 4.

- How the buffer is used in the Algorithm? It is not clear

- Figure 2 is not a clear way to show the summary of the results. Maybe a table is better since now the difference is not clear. Now the question is why default at some cases has the same performance as DLM (no reflection)?
Would not it because of the prompt?

- Why some methods like CoT is not used as a baseline? Or ReAct or Reflexion?

**Limitations:**

It still needs human prompt at the algorithm which make it less practical in terms of type of the community.

The amount of its automation is not very clear. It is good to add it to the discussion.

They need to compare more baselines.

---

> ### Author Rebuttal · Authors · 2024-08-06
>
> 1) “The related works section is not comprehensive…”
>
> We thank the reviewer for the comment. There is growing research in LLMs for healthcare (see Sec. 1). In particular, a question summarization framework in healthcare domains has been proposed [1, 2]. Methods based on contrastive language image pretraining (CLIP) and LLMs to identify medical disorders, generate relevant context, and craft summarizes with visual information have been studied [3, 4]. ChatGPT applications in medical, dental, pharmacy, and public health have been investigated [5, 6]. However, the potential of LLMs to dynamically adapt resource allocation using language prompts, potentially enabling automated policy tuning using expert and community health insights, remains unstudied (see Sec. 1). We will highlight healthcare related works in the paper.
>
> 2) Clarify "policy and critic", "features (z)", and "buffer"
>
> We describe the details of actor-critic algorithms in Appendix F. To improve clarity, we will move this explanation to earlier in Section 3.  Our novel reward proposal loop enhances LLM-generated reward functions using RMAB simulation feedback, and is compatible with various policy learning algorithms, including the widely used actor-critic algorithms. The features describe demographic information such as age range, education level, and income; examples of features (z) are shown in Sec. 4.2, Sec. 4.4 and Sec. 5. We will provide examples earlier at the beginning of Section 4. We train each policy network using RMAB simulations following the concept of experience replay buffers first introduced by [7], widely used to train RL agents using previously observed transitions [8]. We record per-arm state, action, reward, next state, and lambda over n_s timesteps (see Alg. 1 line 11), and store these “trajectories” of arms in the buffer D, which is then used to update our actor-critic algorithms (see Alg. 1 line 13, App. F).
>
> 3) “The DLM with reflection works better, but the comparison is not a good fair comparison. Justifying why without reflection works good or bad is necessary.”
>
> We appreciate the reviewer’s feedback here, but are unclear on why the comparison appears unfair. Could the reviewer clarify their concerns? We refer to DLM (No Reflection) as DLM-NR below. As highlighted (see Sec. 4.5, App. C), LLMs are trained on real-world data, and can therefore recognize and retrieve features relevant to natural language prompts. This is confirmed by our empirical findings: DLM-NR outperforms Default reward policy (see Fig. 2 and Sec. 5.4b), achieving a mean normalized reward (MNR) score of 0.87 +- 0.008 vs. Default score 0.57 +-0.027, with DLM-NR significantly outperforming Default in 11/16 tasks (see Sec. 5.4c). We justify the strong performance of DLM-NR in Table 1, demonstrating more than 70% feature recall in 9/16 tasks; thus, while DLM with Reflection potentially helps fine-tune feature selection and usage (see Fig. 2), DLM-NR still captures relevant features for reward functions without reflection.
>
> 4) “Figure 2 does not clearly summarize results… not clear why Default method sometimes matches the performance of DLM (no reflection).”
>
> We present numerical results in Tab. 4, and observe that DLM-NR significantly outperforms Default in 11/16 tasks. In the remaining tasks, while DLM-NR still performs well in feature selection for reward proposals (see Tab. 1), reflection iterations may be required to fine-tune the usage of these features (see Fig. 3). In some cases, a prompt may have a ground truth Base reward that is close to the original Default reward, yielding higher Default performance; yet even in these cases (see Tab. 4, Idx. 2 “Hindi Speakers”) we find that DLM (Reflection) can help fine-tune features and feature weights to improve over DLM-NR and Default.
>
> 5) “The baselines are not fair … [why not use] CoT, ReAct, or Reflexion”
>
> We note that DLM is a full framework for the use of LLMs in the reward design of an RMAB.  Other cited methods such as  Chain-of-Thought Reasoning (CoT) improve base LLM reasoning and, unlike our proposed model, are not a full framework. The listed methods are in fact compatible to use with our framework; we indeed attempted using CoT, but found it did not improve DLM reasoning by any significant margin (see attached PDF for results). We further emphasize that we compare against random (Fig. 2 red line) and no-action (Fig. 2 purple bar) allocation policy baselines. Thus, we compare against five other baselines in total. ReAct and Reflexion frameworks do not specify how to choose the state space, action space, and reward function for the LLM, which are all significant design decisions that DLM solves. It is unclear how they would be directly comparable to our multi-agent RMAB reward design loop.
>
> 6) “More baseline comparisons are needed”
>
> See (5). We respectfully disagree. As no existing RMAB reward design loop exists before our method, all other comparisons would be additions on top of the proposed DLM framework. We also would emphasize to clarify that in Figure 2, our main results, we indeed compare against 5 other baselines:“No Action”, “Random”, “No Reflection”, “Base”, and “Default”. We clarify that the “Random” baseline is the red line in Fig. 2.
>
> 7) “The level of automation is not clear… the need for human prompts reduces its practicality”
>
> The level of automation is akin to interacting with a chatbot, requiring only prompting and evaluation. Our method is significantly more automated than human design, allowing tasks to be specified via natural language. This significantly reduces the burden that nonprofits face supporting low-resource communities by enabling rapid policy adaptability without significant additional human effort. Note: in an actual deployment (see Sec. 5.5a, Sec. 5.6 and global comment above) a human operator would approve DLM-generated policy outcomes using output state-feature distributions, providing an additional layer of system scrutiny and safety.

---

> > ### Comment · Reviewer_8Afe · 2024-08-08
> >
> > Thanks for the authors to respond my questions.
> > 1- They need to be added in the paper.
> > 2- I still think feature z is unclear. there should a clear and separate part for that.
> > 3- that is what you explained for number 5 and 6.
> > 4- The explanation for the result should be added in the paper to clarify in which cases this algorithm has limitations.
> > 7- The system depends on the prompt. If the prompt is not designed well then the reward is not accurate and consequently the response is not correct.
> > 5, 6- I strongly believe the baselines are not aligned with the claim of the paper. If the claim of the paper is the first time that LLM is used in Healthcare then the results of the sota of methods using LLM should also be in the paper as a reference. It is still not clear how much novelty of the method is worth with respect to the exciting methods.
> > For methods like react, reflexion, the states could be the same as the states used for the current method. They can similarly work with these methods as well. There are many works that used these algorithms in interactive/RL setting including.

---

> > > ### Author Response · Authors · 2024-08-10
> > >
> > > We thank the reviewer for their response and feedback. We agree with the reviewer’s first point, and will include and emphasize all of the above descriptions in the main paper. Specifically, we will include the additional references to work in LLMs for healthcare, and clarify the description of features (z) in a separate section emphasized with a subsection heading. We will additionally emphasize the distinction in performances between Default, DLM-NR, and DLM (Reflection), including cases where Default performance is close to DLM-NR. We note here (point 4) that a higher Default performance does not necessarily imply poorer system performance; rather, this occurs when the Base policy is aligned closely with the Default reward. We find that, even in these cases whether Default reward performs well for a new prompt, we are still able to improve upon reward function proposals through the reflection procedure.
> > >
> > > We also thank the reviewer for their response regarding baseline comparisons. We would like to clarify that our claim is not that we provide “the first time that LLM is used in Healthcare”, but rather “the first to propose using LLMs to adapt to changing resource allocation objectives in public health through _reward design in the RMAB setting_” as stated in the contributions in Section 1. Our goal is _not_ to find how best to use LLMs in this setting, but rather to show that it is possible in the first place, _particularly in the new multi-agent RMAB setting_. This motivates comparison against baselines from the RMAB setting, such as Default and Base rewards, rather than extensive analysis of the best LLM method. We note that adapting single-agent LLM methods like ReAct and Reflexion to the multi-agent RMAB setting is nontrivial. This requires designing a feedback mechanism using multi-agent simulations to guide LLM reward function proposals via state-feature distributions, and dually enabling LLM-generated control in the RMAB setting. _The design of these elements, such as the novel feedback mechanism, is one of the primary contributions of our method, rather than improving LLM reasoning agents such as through ReAct and Reflexion_. As RMAB planners solve a complex combinatorial optimization problem and are specialized for resource allocation, we make the intentional design choice to invoke a separate RMAB planner as a tool to verify outcomes and provide feedback, the key novelty of our framework. This avoids the risks involved with LLMs taking direct actions, especially in the healthcare setting; we instead use LLMs to solve the broader challenge involving natural language and language feedback.
> > >
> > > _Encouraged by the reviewer feedback_, we also propose to add two additional baselines. The first we highlight above (attached PDF), containing additional experiments with chain-of-thought (CoT) reasoning as mentioned in the original review. While we do not find that CoT improved the DLM reasoning in our setting by any significant margin, the results highlight that the listed method of CoT is indeed compatible with our system. The second baseline we propose is an additional noisy-expert baseline, a perturbed Base reward intended to evaluate how an imperfect operator may perform in reward design given a language prompt. This perturbed-Base reward baseline also serves to demonstrate that the problem of coefficient selection in reward function design is indeed non-trivial. We find that this noisy-expert baseline achieves a mean normalized reward (MNR) of 0.87 +- 0.006, compared to DLM (No Reflection) 0.85 +- 0.008 and DLM (Reflection) 0.92 +- 0.006, demonstrating that our method achieves comparable performance to a noisy-expert designer _zeroshot_, and can improve upon zeroshot proposals effectively with the proposed reward reflection module.

---

> > > > ### Comment · Reviewer_8Afe · 2024-08-12
> > > >
> > > > Why then "the first to propose using LLMs to adapt to changing resource allocation objectives in public health through reward design in the RMAB setting" matters in terms of other existing work that have some similar component as the proposed algorithm has?
> > > >
> > > > If the goals is: "show that it is possible in the first place, particularly in the new multi-agent RMAB setting" as you said above then why did you select it to show as the first place? why did you select  multi-agent RMAB setting? What are the reasoning?
> > > >
> > > > I value this work as a contribution to the application of ml algorithm (here RL with reward shaping using LLM) but as a contribution to a top tier conference in ML, there should solid reasoning in why this algorithm is worth and others are not! I did not find this in the paper.  Just RMAB setting, such as Default and Base rewards are giving in the paper which is not enough to show why!
> > > >
> > > > I am not saying you have to use ReAct and Reflexion, I am just suggesting them. You can find any other good compatible algorithm to compare with. Moreover you do not have  to setup  ReAct and Reflexion in MAB, you could just try them as it is, e.g. single agent. It depends on the creativity of authors to set up an existing algorithm to their own idea. Saying simply adapting them is non trivial is not convincing.
> > > >
> > > > Additionally, having your initial results for CoT even shows more that how much selecting fair baseline matters.
> > > >
> > > > I still believe this works has lack of good comparisons and reasoning.

---

> > > > > ### Author Response · Authors · 2024-08-13
> > > > >
> > > > > We sincerely thank the reviewer for their thoughtful and detailed feedback, and for recognizing the contribution of our method and its application in our health setting. We also thank the reviewer for their acknowledgement of our added chain-of-thought baseline above. We appreciate the time the reviewer has taken to engage in this discussion, which has provided valuable clarity and insight. _We understand the reviewer’s perspective regarding the need for more explicit reasoning behind our baseline decisions and the potential for broader comparisons_.
> > > > >
> > > > > We would like to answer the questions raised by the reviewer, and in doing so highlight two points that may provide additional reasoning for our baseline decisions. **First**, we consider the multi-agent RMAB setting in our work for its widespread use in real-world public health resource allocation tasks. Currently, across this setting, the _only_ way to adapt RMAB allocation policies dynamically is via manual policy shaping from human operators. Thus, we consider a manual policy design (our “Base” reward) as the topline or ground-truth performance in our setting; we feel that this is the strongest possible alternative “algorithm” for our method.   **Second**, we clarify that by “adapting single-agent LLM methods (e.g. ReAct and Reflexion) is non-trivial,” we mean to say our key novelty lies in the “translation” between the multi-agent RMAB setting and LLMs: incorporating multi-agent RMAB simulations to guide LLM proposals, and dually leveraging LLMs to adapt RMAB policies dynamically. Thus, comparing alternative LLM techniques (e.g. ReAct and Reflexion) would require us to similarly introduce this “translation” for the multi-agent RMAB setting, which is ultimately one of the key contributions of our work. That is why we consider the listed alternative methods as extensions of LLM reasoning, _rather than_ as direct comparisons to our “translation” framework. As we are the first to propose this “translation” framework to multi-agent RMABs, we instead compare against the strongest existing baseline for shaping policies in real-world RMAB deployments, which is having a manual operator hand-tune policies (e.g.“Base” reward as topline and “noisy-expert” shared above).
> > > > >
> > > > > We greatly appreciate and acknowledge the reviewer’s suggestions above; to improve the reasoning behind the comparisons in our work, we will make the following changes in our paper. First, we will add the chain-of-thought (CoT) experiments and noisy-expert experiments, shared in the attached PDF and above comments, to our paper to provide additional points of comparison for our technique. Second, we will incorporate the above discussion to provide additional reasoning behind our baseline choices, including added detail on the current state-of-the-art in deployed RMAB settings for public health. Third, inspired by the reviewer feedback, we will include a discussion on potential future comparisons that could further enhance the integration of multi-agent RMABs with LLMs, including strategies that may help improve LLM reasoning. We thank the reviewer again for their time and valuable feedback which will help us strengthen our work.

---

### Author Rebuttal · Authors · 2024-08-06

**Thank You**

We thank the reviewers for their insightful feedback and comments. We are encouraged to find that the reviewers recognized the paper as novel, introducing "a new method in this application" (R1), and specifically highlighting its “impressive originality” (R3) and “pioneering” (R1) use of Large Language Models (LLMs) with Restless Multi-Armed Bandits (RMABs) in the public health setting, described as "a clever idea that bridges advanced AI techniques and practical decision-making in a unique way" (R3). We are also grateful that the reviewers appreciated the evaluation of our method in "a simulation environment developed from a real-world dataset," adding credibility to our findings (R2, R3). We are pleased to hear that the reviewers found the paper "very well written and fluent" (R1), with effective use of tables and figures (R3). We also appreciate that the described methodology was noted as aiding reproducibility, including that "the author provides the parameters of the model training process, which helps other researchers reproduce the process" (R2).

**Note on Ethical Considerations**

We appreciate the reviewers raising important points regarding ethical considerations of AI solutions in data consent and privacy. We would like to highlight key sections from the paper that describe our deep consideration of these ethical guidelines during the design process of this work. All of the data collected for use in our _simulated_ public health setting was gathered in close collaboration with our partner NGO, following strict dataset anonymity guidelines (see Sec. 5.1, App. B) and receiving full consent for data collection from each ARMMAN study participant (see App. C.2). In our study, we conduct a _secondary_ analysis of this dataset, representing _simulated_ transition dynamics (see Sec. 5.1, App. C.1). ARMMAN retains full ownership and control over all data; we access _only_ an anonymized version of the data through clearly defined exchange protocols including anonymization, read-only researcher access, and restricted use of data for our secondary analysis (see App. C.2), which was approved by the ARMMAN Board of Ethics (see App. C), registered with the Indian Council of Medical Research (ICMR).

With respect to the use of AI for allocation of resources in _any future deployment_, we note that the proposed system is intended for use _only_ in cases where separate, _additional_ live call resources are available to help underrepresented groups (see Sec. 5.6, App. C.3). These additional live calls provide motivation for users to listen to information in the original voice calls; they do _not_ provide any new information. We therefore do _not_ withhold any original voice call resources through our system, and consider _only_ the additional resources available for the specific use case of dynamic allocation to underrepresented groups as desired by the NGO. This ensures that all the original health voice messages are always available to all participants in the program in any future deployment scenarios (App. C.3). The ARMMAN ethics board has previously approved _deployed_ studies testing RMAB policies for these described additional call resources \[18\], providing evidence of approval of such AI allocation policies (Sec. 1); however, in these cases the RMAB objective was fixed with hand-set objectives, rather than the adaptive policy design we present.

We further note: we do _not_ deploy the proposed method to the ARMMAN program. In any envisioned deployment, we expect that health experts can monitor outcome state-feature distributions of proposed policies, allowing an ethics board to decide whether to adopt system policy suggestions (see Sec. 5.6). Additionally, we note that any potential deployment would follow an extensive real-world evaluation undertaken after securing approval from the ARMMAN Board of Ethics (see Sec. 5.1). Recognizing the significant effort required from the NGO to obtain these approvals and design and conduct such studies, we follow standard deployment protocol to proceed with field trials _only_ when simulations indicate substantial utility to our NGO partner and their ethics board, as demonstrated in this paper. We will ensure that these considerations are emphasized in the paper.

**Response Citations**

[1] Lu et al., "Medical Question Summarization...," ACM TALLIP, 2024.

[2] Caciularu et al., "Long Context Question Answering...," NAACL, 2022.

[3] Ghosh et al., "Clipsyntel: clip and LLM synergy...," AAAI, 2024.

[4] Tiwari et al., "Dr. can see...," CIKM, 2022.

[5] Sallam et al., "ChatGPT applications in medical...," Narra J, 2023.

[6] Fatani, "ChatGPT for future medical...," Cureus, 2023.

[7] Lin, "Self-improving reactive agents...," Machine learning, 1992.

[8] Zhang & Sutton, "A deeper look at experience replay...," arXiv, 2017.

[9] Killian et al., "Restless and uncertain...," UAI, 2022.

[10] Bang et al., "A multitask, multilingual, multimodal evaluation...," arXiv, 2023.

[11] Singh et al., "IndicGenBench: A Multilingual Benchmark...," arXiv, 2024.

[12] WHO, "Social Determinants of Health."

[13] OpenAI, "Prompt Engineering."

[14] OpenAI, "Related Resources Around the Web."

[15] Cole et al., "Selectively answering ambiguous questions...," arXiv, 2023.

[16] Keyvan & Huang, "How to approach ambiguous queries...," ACM CS, 2022.

[17] Min et al., "AmbigQA: Answering ambiguous open-domain questions...," arXiv, 2020.

[18] Verma et al., "Restless Multi-Armed Bandits...," AAMAS, 2023.

---

### Comment · Area_Chair_MUDA · 2024-08-08
**Reviewer - Author Discussion**

Thanks everyone for their hard work on the papers, reviews, and rebuttals. We now have a comprehensive rebuttal from the authors which responds both overall and to each review.

I'd please ask the reviewers to please post a comment acknowledging that they have read the response and ask any followup questions (if any).

This period is to be a discussion between authors and reviewers (Aug 7 - Aug 13) so please do engage now, early in the window, so there is time for a back and forth.

Thanks!

---

### Decision · Program_Chairs · 2024-09-25

**Decision:**

Accept (poster)

**Comment:**

After a review/rebuttal/discussion phase that involved the authors and all reviewers we have come to the recommendation of accepting this paper for NeurIPS.

While there were some ethical concerns, for which the authors received an ethical review whose recommendations need to be followed for the final versions of the paper, the overall consensus for this paper was positive.

In general the reviewers agree that the paper is very well written and clear, the questions are novel and a good application of AI in health care, and the experiments (though limited by only simulation data) are compelling and well presented.